# Align and Distill: Unifying and Improving Domain Adaptive Object Detection

**Justin Kay**[1], **Timm Haucke**[1], **Suzanne Stathatos**[2], **Siqi Deng**[3,*], **Erik Young**[4], **Pietro Perona**[2,3],
**Sara Beery**[1,=], **Grant Van Horn**[5,=]

[1]*MIT,* [2]*Caltech,* [3]*AWS,* [4]*Skagit Fisheries Enhancement Group,* [5]*UMass Amherst*          *kayj@mit.edu*
[=]*Equal advising contribution,* [*]*Work done outside AWS*

**Reviewed on OpenReview:** *https://openreview.net/forum?id=ssXSrZ94sR*

## Abstract

Object detectors often perform poorly on data that differs from their training set. Domain adaptive object detection (DAOD) methods have recently demonstrated strong results on addressing this challenge. Unfortunately, we identify systemic benchmarking pitfalls that call past results into question and hamper further progress: (a) Overestimation of performance due to underpowered baselines, (b) Inconsistent implementation practices preventing transparent comparisons of methods, and (c) Lack of generality due to outdated backbones and lack of diversity in benchmarks. We address these problems by introducing: (1) A unified benchmarking and implementation framework, Align and Distill (ALDI), enabling comparison of DAOD methods and supporting future development, (2) A fair and modern training and evaluation protocol for DAOD that addresses benchmarking pitfalls, (3) A new DAOD benchmark dataset, CFC-DAOD, increasing the diversity of available DAOD benchmarks, and (4) A new method, ALDI++, that achieves state-of-the-art results by a large margin. ALDI++ outperforms the previous state-of-the-art by +3.5 AP50 on Cityscapes → Foggy Cityscapes, +5.7 AP50 on Sim10k → Cityscapes (where ours is the *only* method to outperform a fair baseline), and +0.6 AP50 on CFC-DAOD. ALDI and ALDI++ are architecture-agnostic, setting a new state-of-the-art for YOLO and DETR-based DAOD as well without additional hyperparameter tuning. Our framework[†], dataset[‡], and method offer a critical reset for DAOD and provide a strong foundation for future research.

## 1 Introduction

**The challenge of DAOD.** Modern object detector performance, though excellent across many benchmarks (Lin et al., 2014; Weinstein et al., 2021b;a; Bondi et al., 2018; Schneider et al., 2018; Rodriguez et al., 2011), often severely degrades when test data exhibits a distribution shift with respect to training data (Oza et al., 2023). For instance, detectors do not generalize well when deployed in new environments in environmental monitoring applications (Kay et al., 2022; Weinstein et al., 2021b). Similarly, models in medical applications perform poorly when deployed in different hospitals or on different hardware than they were trained (Xue et al., 2023; Guan & Liu, 2021). Unfortunately, in real-world applications it is often difficult, expensive, or time-consuming to collect the additional annotations needed to address such distribution shifts in a supervised manner. An appealing option in these scenarios is *unsupervised domain adaptive object detection (DAOD)*, which attempts to improve detection performance when moving from a "source" domain (used for training) to a "target" domain (used for testing) (Koh et al., 2021; Kalluri et al., 2023) without the use of target-domain supervision.

**The current paradigm.** The research community has established a set of standard benchmark datasets and methodologies that capture the deployment challenges motivating DAOD. Benchmarks consist of labeled data that is divided into two sets: a source and a target, each originating from different domains. DAOD methods are trained with source-domain images and labels, as in traditional supervised learning, and have access to unlabeled target domain images. The target-domain labels are not available for training.

---

[†] github.com/justinkay/aldi   [‡] github.com/visipedia/caltech-fish-counting

To measure DAOD methods' performance, researchers use *source-only models* and *oracle models* as points of reference. Source-only models—sometimes also referred to as *baselines*—are trained with source-domain data only, representing a lower bound for performance without domain adaptation. Oracle models are trained with *supervised* target-domain data, representing a fully-supervised upper bound. The goal in DAOD is to close the gap between source-only and oracle performance without target-domain supervision.

**Impediments to progress.** Recently-published results indicate DAOD is exceptionally effective, doubling the performance of source-only models and even outperforming fully-supervised oracles (Li et al., 2022b; Chen et al., 2022; Cao et al., 2023). However, upon close examination we discover problems with current benchmarking practices that call these results into question:

P1: Improperly constructed source-only and oracle models, leading to overestimation of performance gains. We find that source-only and oracle models are consistently constructed in a way that does not properly isolate domain-adaptation-specific components, leading to misattribution of performance improvements. We show that when source-only and oracle models are fairly constructed—*i.e.* use the same architecture and training settings as DAOD methods—no existing methods outperform oracles and *many methods do not even outperform source-only models* (Fig. 1), in stark contrast to claims made by recent work. These results mean we do not have an accurate measure of the efficacy of DAOD.

P2: Inconsistent implementation practices preventing transparent comparisons of methods. We find existing DAOD methods are built using a variety of different object detection libraries with inconsistent training settings, making it difficult to determine whether performance improvements come from new DAOD methods or simply improved hyperparameters. We find that tweaking these hyperparameters—whose values often differ between methods yet are not reported in papers—can lead to a *larger change in performance than the proposed methods themselves* (see Section 6.3), thus we cannot take reported advancements at face value. Without the ability to make fair comparisons we cannot transparently evaluate contributions nor make principled methodological progress.

P3: (a) Lack of diverse benchmarks and (b) outdated backbone architectures, leading to overestimation of methods' generality. DAOD benchmarks have focused largely on urban driving scenarios with synthetic distribution shifts (Sakaridis et al., 2018; Johnson-Roberson et al., 2016), and methods continue to use outdated detector backbones for comparison with prior work (Chen et al., 2018). We show that in fact the *ranking of methods changes across benchmarks and architectures*, revealing that published results may be uninformative for practitioners using modern architectures and real-world data.

**A critical reset for DAOD research.** DAOD has the potential for impact in a range of real-world applications, but these systemic benchmarking pitfalls impede progress. We aim to address these problems and lay a solid foundation for future progress in DAOD with the following contributions:

1. *Align and Distill (ALDI)*, a unified benchmarking and implementation framework for DAOD. In order to enable fair comparisons, we first identify key themes in prior work (Section 2) and unify common components into a single state-of-the-art framework, *ALDI* (Section 3). ALDI facilitates detailed study of prior art and streamlined implementation of new methods, supporting future research.

2. A fair and modern training protocol for DAOD methods, enabled by ALDI. We provide quantitative evidence of the benchmarking pitfalls we identify and propose an updated training and evaluation protocol to address them (Section 6.1). This enables us to set more realistic and challenging targets for the DAOD community and perform the first fair comparison of prior work in DAOD (Section 6.2).

3. A new benchmark dataset, CFC-DAOD, sourced from a real-world adaptation challenge in environmental monitoring (Section 5). CFC-DAOD increases the diversity of DAOD benchmarks and is notably larger than existing options. We show that the ranking of methods changes across different benchmarks (Section 6.2), thus the community will benefit from an additional point of comparison.

4. A new method, ALDI++, that achieves state-of-the-art results by a large margin. Using the same model settings across all benchmarks, ALDI++ outperforms the previous state-of-the-art by +3.5 AP50 on Cityscapes $\rightarrow$ Foggy Cityscapes, +5.7 AP50 on Sim10k $\rightarrow$ Cityscapes (where ours is the *only* method to outperform a fair source-only model), and +2.0 AP50 on CFC Kenai $\rightarrow$ Channel.

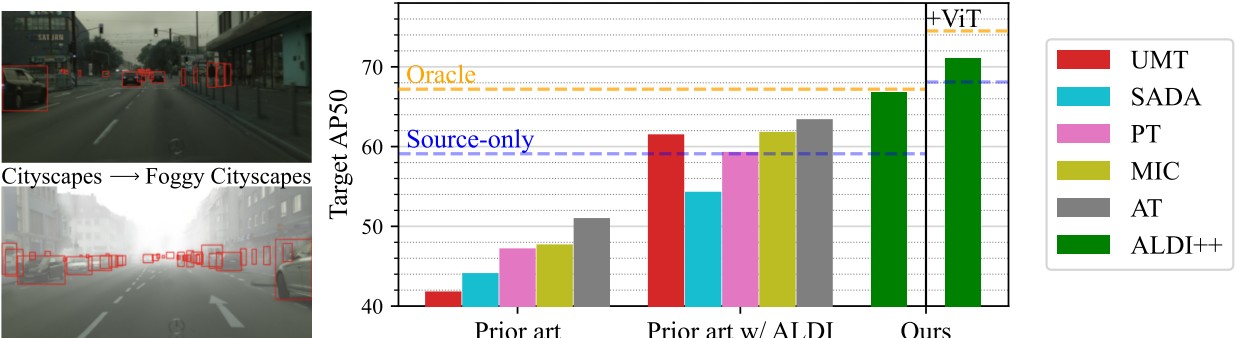

Figure 1: **ALDI provides a unified framework for fair comparison of domain adaptive object detection (DAOD) methods, and ALDI++ achieves state-of-the-art performance.** We show: **(1)** Inconsistent implementation practices give the appearance of steady progress in DAOD (left bars (Deng et al., 2021; Chen et al., 2021; 2022; Hoyer et al., 2023; Li et al., 2022b)); reimplementation and fair comparison with ALDI shows less difference between methods than previously reported (middle bars); **(2)** A fairly constructed source-only model (blue line) outperforms many existing DAOD methods, indicating less progress has been made than previously reported; and a proper oracle (orange line) outperforms *all* existing methods, in contrast to previously-published results; and **(3)** Our proposed method ALDI++ (green bars) achieves state-of-the-art performance on DAOD benchmarks such as Cityscapes → Foggy Cityscapes and is complementary to ongoing advances in object detection like VitDet (Li et al., 2022a).

## 2 Related Work

Our work concerns domain adaptive 2D object detection (DAOD). Two methodological themes have dominated recent DAOD research: *feature alignment* and *self-training/self-distillation*. We first give an overview of these themes and previous efforts to combine them, and then use commonalities to motivate our unified framework, *Align and Distill*, in Section 3.

**Feature alignment in DAOD.** Feature alignment methods aim to make target-domain data "look like" source-domain data, reducing the magnitude of the distribution shift. The most common approach utilizes an adversarial learning objective to align the feature spaces of source and target data (Ganin & Lempitsky, 2015; Chen et al., 2021; 2018; Zhu et al., 2019). Faster R-CNN in the Wild (Chen et al., 2018) utilizes adversarial networks at the image and instance level. SADA (Chen et al., 2021) extends this to multiple adversarial networks at different feature levels. Other approaches propose mining for discriminative regions (Zhu et al., 2019), weighting local and global features differently (Saito et al., 2019), incorporating uncertainty (Nguyen et al., 2020), and using attention networks (Vs et al., 2021). Alignment at the pixel level has also been proposed using image-to-image translation techniques to modify input images directly (Deng et al., 2021).

**Self-training/self-distillation in DAOD.** Self-training methods use a "teacher" model to predict pseudo-labels on target-domain data that are then used as training targets for a "student" model. Self-training can be seen as a type of *self-distillation* (Pham et al., 2022; Caron et al., 2021), which is a special case of knowledge distillation (Hinton et al., 2015; Chen et al., 2017) where the teacher and student models share the same architecture. Most recent self-training approaches in DAOD are based on the Mean Teacher (Tarvainen & Valpola, 2017) framework, in which the teacher model is updated as an exponential moving average (EMA) of the student model's parameters. Extensions to Mean Teacher for DAOD include: MTOR, which utilizes graph structure to enforce student-teacher feature consistency (Cai et al., 2019), Probabilistic Teacher (PT), which uses probabilistic localization prediction and soft distillation losses (Chen et al., 2022), and Contrastive Mean Teacher (CMT), which uses MoCo (He et al., 2020) for student-teacher consistency (Cao et al., 2023).

**Combining feature alignment and self-training.** Several approaches utilize *both* feature alignment and self-training/self-distillation, motivating our unified framework. Unbiased Mean Teacher (UMT) (Deng et al., 2021) uses mean teacher in combination with image-to-image translation to align source and target data at

the pixel level. Adaptive Teacher (AT) (Xue et al., 2023) uses mean teacher with an image-level discriminator network. Masked Image Consistency (MIC) (Hoyer et al., 2023) uses mean teacher, SADA, and a masking augmentation to enforce teacher-student consistency. Because these methods were implemented in different codebases using different training recipes and hyperparameter settings, it is unclear which contributions are most effective and to what extent feature alignment and self-training are complementary. We address these issues by reimplementing these approaches in the ALDI framework and perform fair comparisons and ablation studies in Section 6.

**DAOD implementations.** There are two components to an object detector design: the detection architecture (*e.g.* Faster R-CNN Ren et al. (2015), YOLO Redmon et al. (2016), DETR Carion et al. (2020)) and the backbone (*e.g.* VGG Simonyan & Zisserman (2014), ResNet He et al. (2016), ViT Dosovitskiy et al. (2020)). Current state-of-the-art methods in DAOD predominantly use Faster R-CNN architectures. DOAD methods for YOLO and DETR backbones have recently received some attention Zhou et al. (2023); Yu et al. (2022); Jia et al. (2023), but have yet to surpass Faster R-CNN-based methods' performance. For this reason, our main experiments also utilize the Faster R-CNN architecture. Existing methods differ in their choice of backbone, making comparisons difficult; we address this by consistently utilizing ResNet-50 backbones for all experiments and in our re-implementations of prior work. However, the ALDI framework is architecture and backbone agnostic, and we provide additional experiments using YOLO and DETR architectures, as well as ViT and ConvNeXt Liu et al. (2022) backbones.

**DAOD datasets.** Cityscapes (CS) → Foggy Cityscapes (FCS) (Cordts et al., 2016; Sakaridis et al., 2018) is a popular DAOD benchmark that emulates domain shift caused by changes in weather in urban driving scenarios. The dataset contains eight vehicle and person classes. Sim10k → CS (Johnson-Roberson et al., 2016) poses a Sim2Real challenge, adapting from video game imagery to real-world imagery. The benchmark focuses on a single class, "car". Other common tasks include adapting from real imagery in PascalVOC (Everingham et al., 2010) to clip art and watercolor imagery (Inoue et al., 2018). We report results on CS → FCS and Sim10k → CS due to their widespread popularity in the DAOD literature and focus on real applications. We note that existing benchmarks reflect a relatively narrow set of potential DAOD applications. To study whether methods generalize outside of urban driving scenarios, in Section 5 we introduce a novel dataset sourced from a real-world adaptation challenge in environmental monitoring, where imagery is much different from existing benchmarks.

## 3 Align and Distill (ALDI): Unifying DAOD

We first introduce *Align and Distill (ALDI)*, a new benchmarking and implementation framework for DAOD. ALDI unifies feature alignment and self-distillation approaches in a common framework, enabling fair comparisons and addressing P2. Inconsistent implementation practices, while also providing the foundation for development of a new method ALDI++ that achieves state-the-art performance (Section 4, Section 6.2). The framework is visualized in Fig. 2. All components are ablated in Section 6.3.

**Data.** DAOD involves two datasets: a labeled source dataset $X_{src}$ and an unlabeled target dataset $X_{tgt}$. Each training step, a minibatch of size $B$ is constructed containing both $B_{src}$ source images and $B_{tgt}$ target images, $B = B_{src} + B_{tgt}$.

**Models.** ALDI is designed as a student-teacher framework to facilitate algorithms utilizing self-training/self-distillation. When enabled, both a student model $\theta_{stu}$ and a teacher model $\theta_{tch}$ are initialized with the same weights, typically obtained through supervised pretraining on ImageNet or $X_{src}$. Pretraining on $X_{src}$ is often referred to as "burn-in." The student is trained via backpropagation, while the teacher's weights are updated each training step to be the EMA of the student's weights (Tarvainen & Valpola, 2017), *i.e.* $\theta_{tch} = \alpha\theta_{tch} + (1 - \alpha)\theta_{stu}$ with $\alpha \in [0, 1]$. After training, we keep $\theta_{tch}$ and discard $\theta_{stu}$. Algorithms that do not use self-training/self-distillation (*e.g.* SADA (Chen et al., 2021)) simply disable $\theta_{tch}$.

In this paper we focus predominantly on two-stage detectors based on Faster R-CNN (Ren et al., 2015) as they are currently the state-of-the-art in DAOD, though we note that our framework is architecture-agnostic and also supports YOLO and DETR-based detectors. We provide additional YOLO and DETR results in Appendix A.1.

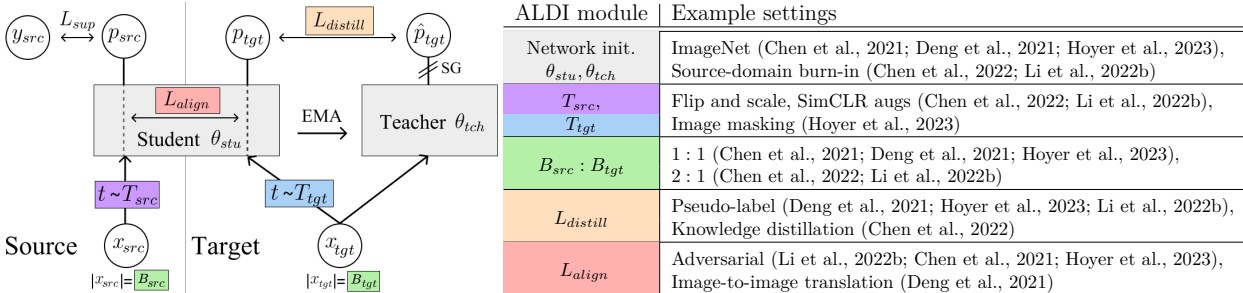

Figure 2: **(Left) The ALDI framework.** Each training step (moving left to right and bottom to top): **(1)** Sample $B_{\mathrm{src}}$ labeled source images $x_{\mathrm{src}}$; transform by $t \sim T_{\mathrm{src}}$; pass to student; compute supervised loss $L_{\mathrm{sup}}$ using ground-truth labels $y_{\mathrm{src}}$. **(2)** Sample $B_{\mathrm{tgt}}$ unlabeled target images $x_{\mathrm{tgt}}$; transform by $t \sim T_{\mathrm{tgt}}$; pass to student to get preds $p_{\mathrm{tgt}}$. Compute alignment objectives $L_{\mathrm{align}}$ using $x_{\mathrm{src}}$ and $x_{\mathrm{tgt}}$. **(3)** Pass same unlabeled target data $x_{\mathrm{tgt}}$, weakly transformed, to teacher; postprocess to obtain teacher predictions $\hat{p}_{\mathrm{tgt}}$. Compute distillation loss $L_{\mathrm{distill}}$ between teacher and student predictions. Use stop gradient (SG) on teacher model; update teacher to the EMA of student's weights. **(Right) Example settings for each component of ALDI.** ALDI supports a range of existing methods off-the-shelf while providing a general implementation framework for new methods.

**Training** involves one or more of the following three objectives. We note that each objective is optional in order to support a range of algorithmic approaches.

**1. Supervised training with source data.** For each labeled source sample $x_{\mathrm{src},i}$, we apply a transformation $t \sim T_{\mathrm{src}}$, where $T_{\mathrm{src}}$ is the set of possible source-domain transformations. The transformed sample is passed through the student model to compute the supervised loss $\mathcal{L}_{\mathrm{sup}}$ given the ground truth targets $y_{\mathrm{src},i}$:

$$\mathcal{L}_{\mathrm{sup}} = \frac{1}{B_{\mathrm{src}}} \sum_{i=1}^{B_{\mathrm{src}}} \mathcal{L}\left(\theta_{\mathrm{stu}}(t(x_{\mathrm{src},i})), y_{\mathrm{src},i}\right) \tag{1}$$

where $\mathcal{L}(\cdot, \cdot)$ are standard object detection loss functions, *e.g.* those of Faster R-CNN (Ren et al., 2015).

**2. Self-distillation with target data.** For each unlabeled target sample $x_{\mathrm{tgt},i}$, we transform the input using $\hat{t} \sim T_{\mathrm{weak}}$ (a set of weak transformations) for the teacher model and $t \sim T_{\mathrm{tgt}}$ (stronger transformations) for the student model. The teacher's predictions $\hat{p}_{\mathrm{tgt},i}$ serve as distillation targets for the student's predictions $p_{\mathrm{tgt},i}$, and we compute distillation loss $\mathcal{L}_{\mathrm{distill}}$ :

$$\hat{p}_{\mathrm{tgt},i} = \theta_{\mathrm{tch}}(\hat{t}(x_{\mathrm{tgt},i})) \quad (2) \qquad p_{\mathrm{tgt},i} = \theta_{\mathrm{stu}}(t(x_{\mathrm{tgt},i})) \quad (3) \qquad \mathcal{L}_{\mathrm{distill}} = \frac{1}{B_{\mathrm{tgt}}} \sum_{i=1}^{B_{\mathrm{tgt}}} \mathcal{L}_{\mathrm{distill}}\left(p_{\mathrm{tgt},i}, \hat{p}_{\mathrm{tgt},i}\right) \quad (4)$$

where teacher outputs $\hat{p}_{\mathrm{tgt},i}$ are postprocessed to be either soft (*e.g.*, logits or softmax outputs) or hard (*e.g.*, thresholded pseudo-label) targets and the choice of distillation loss is method-specific. This formulation unifies different distillation techniques into a common objective, supporting a range of approaches.

**3. Feature alignment.** The source samples $x_{\mathrm{src},i}$ and target samples $x_{\mathrm{tgt},i}$ are optionally "aligned" using an alignment objective $\mathcal{L}_{\mathrm{align}}$ that enforces invariance across domains at either the image or feature level. This formulation is general; however, in this paper, we focus on two common alignment losses: domain-adversarial training and image-to-image alignment.

Domain-adversarial training (*i.e.* DANN (Ganin & Lempitsky, 2015)) trains a domain classifier $D$ to distinguish between source and target features, while the feature extractor aims to confuse $D$:

$$\mathcal{L}_{\mathrm{align,DANN}} = -\frac{1}{B} \sum_{i=1}^{B} \left[y_{\mathrm{dom},i} \log(D(\theta(x_i))) + (1 - y_{\mathrm{dom},i}) \log(1 - D(\theta(x_i)))\right] \tag{5}$$

where $y_{\text{dom},i}$ is the domain label (source = 0, target = 1) and $\theta(x_i)$ is a feature representation of $x_i$.

Image-to-image alignment instead pursues domain invariance in the pixel space. Given image-to-image generative models $G_{\text{src}}$, $G_{\text{tgt}}$ (*e.g.*, a CycleGAN (Zhu et al., 2017)), images are "translated" (a pixel-level transformation) from the source domain to the target domain and vice versa. We then obtain $x_{\text{tgt-like},i} = G_{\text{src}}(x_{\text{src},i})$, $x_{\text{src-like},i} = G_{\text{tgt}}(x_{\text{tgt},i})$, and substitute into Eq. (1), Eq. (2), and Eq. (3).

**Unification of prior work.** We demonstrate the generality of our framework by reimplementing five recently-proposed methods on top of ALDI for fair comparison: UMT (Deng et al., 2021), SADA (Chen et al., 2021), PT (Chen et al., 2022), MIC (Hoyer et al., 2023), and AT (Li et al., 2022b). We enumerate the settings required to reproduce each method in Appendix C.

## 4 ALDI++: Improving DAOD

We next propose a set of simple but effective enhancements to the *Align and Distill* approach. We call the resulting method ALDI++. We show in Section 6.2 that these enhancements lead to state-of-the-art results, and ablate each component in Section 6.3.

**1. Robust burn-in.** A key challenge in student-teacher methods is improving target-domain pseudo-label quality. We point out that pseudo-label quality in the early stages of self-training is largely determined by the *out-of-distribution (OOD) generalization* capabilities of the initial teacher model $\theta_{tch}^{init}$, and thus propose a pre-training ("burn-in") strategy aimed at improving OOD generalization *before* self-training.

We add strong data augmentations including random resizing, color jitter, and cutout (DeVries & Taylor, 2017; Chen et al., 2020), and keep an EMA copy of the model during burn-in, two strategies that have previously been shown to improve OOD generalization (Morales-Brotons et al., 2024; Arpit et al., 2022; Gao et al., 2022), *i.e.* we pre-train a model $\theta$ with the loss from Eq. (1), where $t \sim \mathcal{T}_{\text{src}}$ and $L_{\text{sup}}$ are still the standard Faster R-CNN losses. Each iteration we update an EMA copy of the model,

$$\theta^{\text{EMA}} = \alpha\theta^{\text{EMA}} + (1 - \alpha)\theta \tag{6}$$

with $\alpha \in [0, 1]$. After pre-training, we initialize $\theta_{\text{stu}} = \theta_{\text{tch}} = \theta^{\text{EMA}}$. We are the first to utilize these strategies for DAOD burn-in, and we show in Section 6.3 that this pre-training strategy leads to faster convergence time and better results.

**2. Multi-task soft distillation.** Most prior work utilizes confidence thresholding and non-maximum suppression to generate "hard" pseudo-labels from teacher predictions $\hat{p}_{tgt}$. However in object detection this strategy is sensitive to the confidence threshold, leading to false positive and false negative errors that harm self-training (Kay et al., 2023; RoyChowdhury et al., 2019). Inspired by the knowledge distillation literature we propose instead using "soft" distillation losses—*i.e.* using teacher prediction scores as targets without thresholding—allowing us to eliminate the confidence threshold hyperparameter.

We describe here our approach for two-stage (Faster R-CNN-based) object detection. Distillation implementation details for YOLO and DETR architectures can be found in Appendix B. We distill each task of Faster R-CNN—Region Proposal Network localization (*rpn*) and objectness (*obj*), and Region-of-Interest Heads localization (*roih*) and classification (*cls*)—independently. At each stage, the teacher provides distillation targets for the same set of input proposals used by the student—*i.e.* anchors $A$ in the first stage, and *student* region proposals $p_{tgt}^{rpn}$ in the second stage:

$$p_{tgt}^{rpn,obj} = \theta_{stu}^{rpn,obj}(A, x_{tgt}^t) \tag{7} \qquad\qquad \hat{p}_{tgt}^{rpn,obj} = \theta_{tch}^{rpn,obj}(A, x_{tgt}^{\hat{t}}) \tag{8}$$

$$p_{tgt}^{roih,cls} = \theta_{stu}^{roih,cls}(p_{tgt}^{rpn}, x_{tgt}^t) \tag{9} \qquad\qquad \hat{p}_{tgt}^{roih,cls} = \theta_{tch}^{roih,cls}(p_{tgt}^{rpn}, x_{tgt}^{\hat{t}}) \tag{10}$$

At each iteration, student distillation losses $L_{distill}$ are computed as:

$$L_{distill}^{rpn} = \lambda_0 L_{rpn}(p_{tgt}^{rpn}, \hat{p}_{tgt}^{rpn}) + \lambda_1 L_{obj}(p_{tgt}^{obj}, \hat{p}_{tgt}^{obj}) \tag{11}$$

$$L_{distill}^{roih} = \lambda_2 L_{roih}(p_{tgt}^{roih}, \hat{p}_{tgt}^{roih}) + \lambda_3 L_{cls}(p_{tgt}^{cls}, \hat{p}_{tgt}^{cls}) \tag{12}$$

$$L_{distill} = L_{distill}^{rpn} + L_{distill}^{roih} \tag{13}$$

Where $L_{rpn}$ and $L_{roih}$ are the smooth L1 loss and $L_{obj}$ and $L_{cls}$ are the cross-entropy loss, and $\lambda_{0\ldots3} = 1$ by default. See Fig. 2 for a visual depiction, and the appendix for implementation details.

One prior DAOD work, PT (Chen et al., 2022), has also used soft distillation losses. Our method addresses two shortcomings: (1) PT requires a custom "Probabilistic R-CNN" architecture for distillation, while our approach is general and can work with any two-stage detector, and (2) PT uses $\hat{p}^{cls}$ as an indirect proxy for distilling $p^{obj}$, while our approach distills each task directly.

**3. Revisiting DAOD training recipes.** We also re-examine common design choices in DAOD in order to establish strong baseline settings for ALDI++. In particular, we find that two simple changes consistently improve domain adaptation results: (1) Using strong regularization on both target *and* source data during self-training, and (2) Training with equal amounts of source and target supervision in each minibatch (*i.e.* $B_{src} = B_{tgt}$). We also opt to disable all feature alignment in ALDI++ to stabilize training and find that the effects on accuracy are minimal (see Section 6.3).

## 5 The CFC-DAOD Dataset

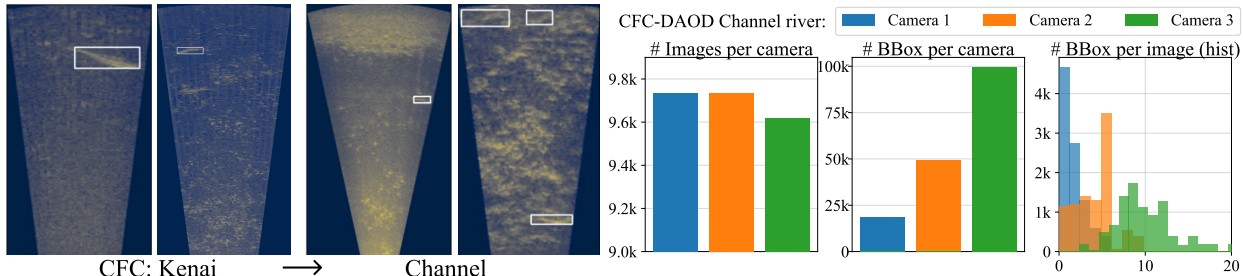

Figure 3: **The CFC-DAOD benchmark** focuses on detecting fish (white bounding boxes) in sonar imagery under domain shift caused by environmental differences between the training location (Kenai) and testing location (Channel). Our dataset contains 168k bounding boxes in 29k frames sampled from 150 new videos captured over two days from 3 different sonar cameras on the Channel river, enabling DAOD experiments. Here we visualize the distribution of images and annotations from each camera.

Next we introduce our dataset contribution, CFC-DAOD, as a step toward addressing P3: (a) Lack of diverse benchmarks leading to overestimation of methods' generality.

**CFC.** The Caltech Fish Counting Dataset (CFC) (Kay et al., 2022) is a *domain generalization* benchmark sourced from fisheries monitoring, where sonar video is used to detect and count migrating salmon. The detection task consists of a single class ("fish") and domain shift is caused by real-world environmental differences between camera deployments. We identify this application as an opportunity to study the generality of DAOD methods due to its stark differences with existing DAOD benchmarks—specifically, sonar imagery is grayscale, has low signal-to-noise ratios, and foreground objects are difficult to distinguish from the background—however CFC focuses on generalization rather than adaptation and *does not include the data needed for DAOD*.

**CFC-DAOD** We introduce an extension to CFC, deemed CFC-DAOD, to enable the study of DAOD in this application domain. The task is to adapt from a source location—"Kenai", *i.e.* the default training set from CFC—to a difficult target location, "Channel". We collected an additional 168k bounding box annotations in 29k frames sampled from 150 new videos captured over two days from 3 different sensors on the "Channel" river (see Fig. 3). For consistency, we closely followed the video sampling protocol used to collect the original CFC dataset as described by the authors (see (Kay et al., 2022)). Our addition to CFC is crucial for DAOD as it adds an unsupervised training set for domain adaptation methods and a supervised training set to train oracle methods. We keep the original supervised Kenai training set from CFC (132k annotations in 70k images) and the original Channel test set (42k annotations in 13k images). We note

this is substantially larger than existing DAOD benchmarks (CS contains 32k instances in 3.5k images, and Sim10k contains 58k instances in 10k images). See Appendix D for more dataset statistics and Appendix G for qualitative visualizations. We make the dataset public.

# 6   Experiments

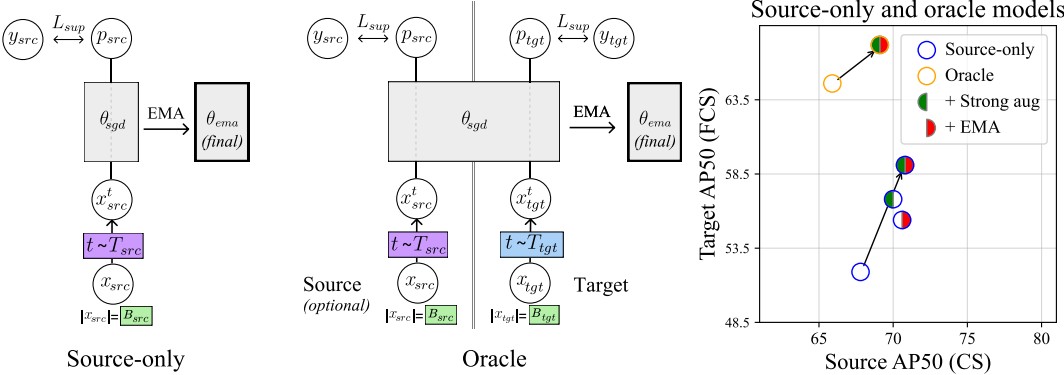

Figure 4: **Revisiting source-only and oracle models in DAOD.** We argue that in order to provide a fair measure of *domain adaptation* performance in DAOD, source-only and oracle models must utilize the same *non-adaptive* architectural and training components as methods being studied. In the case of *Align and Distill*-based approaches, this means source-only and oracle models must have access to the same set of source augmentations and EMA as DAOD methods. We see that these upgrades significantly improve source-only performance on target-domain data (+7.2 AP50 on Foggy Cityscapes), *even though the source-only model has never seen any target-domain data*, and these upgrades also improve oracle performance. Overall, these results set more challenging and realistic performance targets for DAOD methods.

In this section we propose an updated benchmarking protocol for DAOD (Section 6.1) that allows us to fairly analyze the performance of ALDI++ compared to prior work (Section 6.2) and conduct extensive ablation studies (Section 6.3).

## 6.1   Benchmarking Protocol

**Datasets.** We perform experiments on Cityscapes → Foggy Cityscapes, Sim10k → Cityscapes, and CFC Kenai → Channel. In addition to being consistent with prior work, these datasets represent three common adaptation scenarios capturing a range of real-world challenges: weather adaptation, Sim2Real, and environmental adaptation, respectively. We note that there have been inconsistencies in prior work in terms of which ground truth labels for Cityscapes are used. We use the Detectron2 version, which includes three intensity levels of fog $\{0.005, 0.01, 0.02\}$.

**Metrics.** For all experiments we report the PascalVOC metric of mean Average Precision with IoU $\geq 0.5$ ("AP50") (Everingham et al., 2010). This is consistent with prior work on Cityscapes, Foggy Cityscapes, Sim10k, and CFC.

**Revisiting source-only and oracle models.** Here we address P1: Improperly constructed source-only and oracle models, leading to overestimation of performance gains. The goal of DAOD is to develop adaptation techniques that use unlabeled target-domain data to improve target-domain performance. Thus, in order to properly isolate *adaptation-specific* techniques, **any technique that does not need target-domain data to run should also be used by source-only and oracle models**. This means that source-only and oracle models should also utilize the same strong augmentations and EMA updates as DAOD methods.

In Fig. 4 we illustrate the resulting source-only and oracle models, and show that including these components significantly improves both source-only and oracle model performance (+7.2 and +2.6 AP50 on Foggy Cityscapes, respectively). This has significant implications for DAOD research: because source-only and

oracle models have not been constructed with equivalent components, performance gains stemming from better generalization have until now been *misattributed* to DAOD. With properly constructed source-only and oracle models, the gains from DAOD are much more modest (see Fig. 5). Note that for clarity we compare all methods in Fig. 1 and Fig. 5 against a single source-only and oracle model, however it would be more appropriate to compare each method to its own bespoke source-only and oracle models that use the same training components; see Appendix A.7 for the full comparison.

**Fixed training settings.** Prior work has used inconsistent backbones and image sizes, making head-to-head comparisons less fair. Using ALDI we instead compare using the same training settings, offering a fair comparison. As a starting point we utilize reasonably modern settings likely to be used by a practitioner: the Cityscapes defaults in the Detectron2 codebase. All methods in our comparisons, including source-only and oracle models, utilize Faster R-CNN architectures with ResNet-50 (Ren et al., 2015) backbones with FPN (Lin et al., 2017), COCO (Lin et al., 2014) pre-training, and an image size of 1024px on the shortest side. See Appendix C for more details.

## 6.2 Fair Comparison and State-of-the-Art Results

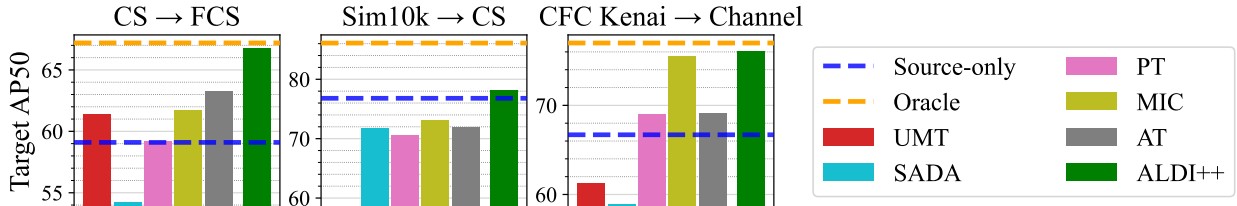

Figure 5: **Fair comparison of ALDI++ with existing state-of-the-art approaches using the ALDI framework and modern training recipes.** Some prior methods show consistent benefit but others lag behind fair source-only models. Our method ALDI++ outperforms prior work on all datasets studied by a significant margin: +3.5 AP50 on CS → FCS, +5.7 AP50 on Sim10k → CS, and +0.6 AP50 on CFC Kenai → Channel. Notably, ALDI++ is the *only* method to outperform a source-only model on Sim10k → CS.

We compare ALDI++ with reimplementations of five state-of-the-art DAOD methods on top of our framework: UMT (Deng et al., 2021), SADA (Chen et al., 2021), PT (Chen et al., 2022), MIC (Hoyer et al., 2023), and AT (Li et al., 2022b); see Appendix C for the ALDI settings used to reproduce them. We use the fair benchmarking protocol proposed in Section 6.1. Results are shown in Fig. 5. All methods (including ALDI++) use the same settings for all benchmarks.

**Comparison with state-of-the-art.** ALDI++ outperforms all prior work and sets a new state-of-the-art on all benchmarks studied, outperforming the next-best methods by +3.5 AP50 on CS → FCS, +5.7 AP50 on Sim10k → CS (where ours is the *only* method to outperform a fair source-only model), and +0.6 AP50 on CFC Kenai → Channel. ALDI++ achieves near-oracle level performance on CS → FCS and CFC Kenai → Channel (0.4 and 0.9 AP50 away, respectively), while other methods close less than half the gap between source-only and oracle models.

**Comparison across datasets.** We compare all methods on CS → FCS, Sim10k → CS, and CFC Kenai → Channel, in Fig. 5. We find the ranking of methods differs across datasets. ALDI++, MIC and AT are consistently the top-performing methods across all datasets. MIC performs noticeably better on CFC Kenai → Channel than other prior work, nearly matching the performance of ALDI++. UMT exhibits variable performance due to the differences in the difficulty of image generation across datasets (see Appendix C for examples). SADA underperforms other methods on CS → FCS and CFC Kenai → Channel, but closes this gap on the more difficult Sim10k → CS. These results demonstrate the utility of CFC-DAOD as another point of comparison for DAOD methods; we see that method performance on synthetic benchmarks like CS → FCS is not necessarily indicative of performance on real-world domain shifts.

**Comparison with fair source-only and oracle models**. Re-implementing methods in ALDI improves absolute performance of most methods due to upgraded training settings; however performance decreases

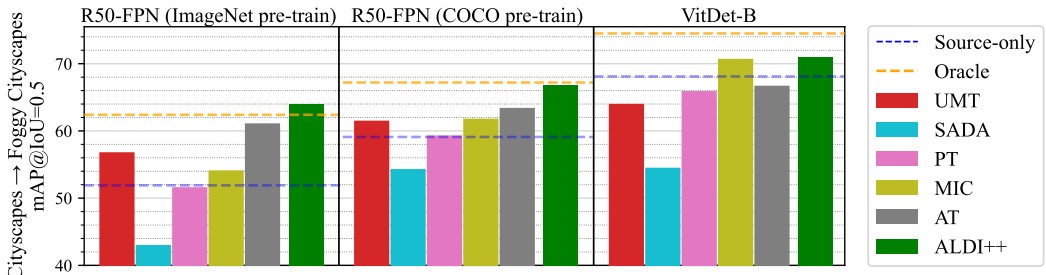

Figure 6: **Comparison across backbones and pre-training data.** Pre-training strategy does not significantly change the relative strength of methods compared to each other nor compared to source-only and oracle models. However, all models perform worse compared to source-only and oracle models when using VitDet backbones, with only ALDI++ and MIC outperforming a source-only model.

dramatically compared to source-only and oracle models. There are several instances where modernized DAOD methods are actually *worse* than a fair source-only model. Notably, a source-only model outperforms upgraded versions *all* previously-published work on Sim10 → CS. We also see that no state-of-the-art methods outperform a fair oracle on any dataset, in contrast to claims made by prior work (Li et al., 2022b; Chen et al., 2022; Cao et al., 2023).

**Comparison across backbones and pre-training data.** We compare all methods on CS → FCS using ImageNet vs. COCO pre-training, as well as Resnet-50 FPN backbones vs. VitDet-B backbones, in Fig. 6. See Appendix F for other datasets. We see that while COCO pre-training improves absolute performance of all methods, their ranking does not change significantly compared to ImageNet pre-training, nor does their performance in relation to source-only baselines. Interestingly, we find that ALDI++ outperforms an oracle on CS → FCS when using ImageNet pre-training, however we do not observe this trend on the other datasets (see Appendix F). We hypothesize that this may be due to noise in the Foggy Cityscapes target-train labels due to their programmatic generation, and that COCO pre-training helps prevent the oracle from overfitting to these erroneous boxes. We show that only ALDI++ and MIC continue to show improvements over an upgraded VitDet source-only model (see Appendix A.2 for other datasets), with ALDI++ performing slightly better than MIC (+0.4 AP50). We see there is a larger gap between the ViT ALDI++ and the ViT oracle compared to ResNet backbones, indicating the potential for future work to improve performance. Across all experiments in Fig. 6 we see that the source-only–oracle gap shrinks as the underlying model improves due to either stronger pre-training or backbone upgrades, indicating that DAOD may offer diminishing returns with stronger models.

## 6.3 Ablation Studies

In this section we ablate the performance of each component of ALDI on CS → FCS.

**Base settings.** For each ablation, unless otherwise specified we begin with the following training settings. We initialize $\theta_{stu}, \theta_{tch}$ with COCO pre-training followed by a burn-in phase on $X_{\text{src}}$ with weak augmentations and early stopping based on validation performance. $T_{\text{src}}$ includes random horizontal flip and random scaling. $T_{\text{tgt}}$ includes random horizontal flip, random scaling, color jitter, and cutout. The $B_{\text{src}} : B_{\text{tgt}}$ batch ratio is 1:1. $L_{\text{distill}}$ is hard pseudo-labeling with a confidence threshold of 0.8, and $L_{\text{align}}$ is disabled. Note these base settings are not necessarily those of ALDI++ but rather the most commonly chosen values in prior work for each component. Additional training settings are reported in Appendix C.

$\theta_{stu}, \theta_{tch}$ **Network initialization (burn-in).** In Fig. 7a we analyze the effects of our proposed burn-in strategy (see Section 4). We measure performance in terms of target-domain AP50 as well as convergence time, defined as the training time at which the model first exceeds 95% of its final target-domain performance. We compare our approach with: (1) No dataset-specific burn-in, *i.e.* starting with COCO weights, and (2) The approach used by past work—using a fixed burn-in duration, *e.g.* 10k iterations. We find that our method results in significant improvements in both training speed and accuracy, leading to upwards of 10% improvements in AP50 and reducing training time by a factor of 10 compared to training without burn-in.

Table 1: **Ablation studies.** **(a)** Effects of target-domain augmentation on self-training. Augmentations applied to student inputs ($T_{tgt}$ in Fig. 2). Stronger augmentations improve performance considerably. **(b)** Effects of distillation objectives on self-training. We compare hard targets—used by most prior art, which thresholds teacher predictions to create pseudo-labels—with our proposed soft targets. Soft targets can improve overall performance. Results are the mean and standard deviation over 3 runs. **(c)** Feature alignment has diminishing returns. Compared to a source-only baseline AP50 of 59.1, feature alignment objectives $L_{align}$ without self-training provides up to 2.6 AP50 of benefit (first row), but diminishes to 0.2 AP50 additional gain when used alongside self-training (last row).

(a)

| $T_{tgt}$ | $AP50_{FCS}$ |
|---|---|
| Weak (scale & flip) | 52.6 |
| + Color jitter | 59.0 |
| + Color jitter + Erase | 63.1 |
| + Color jitter + MIC | 64.3 |

(b)

| $L_{distill}$ | $AP50_{FCS}$ |
|---|---|
| Hard targets | $63.7 \pm 0.1$ |
| Soft targets | $64.0 \pm 0.4$ |

(c)

| $L_{align}$ | $L_{distill}$ | $AP50_{FCS}$ |
|---|---|---|
| ✓ | | 61.7 |
| | ✓ | 63.7 |
| ✓ | ✓ | 63.9 |

$T_{src}$ **Source augmentations.** In Fig. 7b we ablate the set of source-domain data augmentations. We compare using weak augmentations (random flipping and random scaling), strong augmentations (color jitter and cutout), and a combination of weak and strong, noting that prior works differ in this regard but do not typically report the settings used. We find that using strong source augmentations on the entire source-domain training batch outperforms weak augmentations and a combination of both.

$T_{tgt}$ **Target augmentations.** In Table 1a we investigate the use of different augmentations for target-domain inputs to the *student* model. (We note that weak augmentations are always used for target-domain inputs to the teacher in accordance with prior work). We see that stronger augmentations consistently improve performance, with best performance coming from the recently-proposed MIC augmentation (Hoyer et al., 2023).

$B_{tgt}/B$ **Batch composition.** In Fig. 7c we ablate the ratio of source and target data within a minibatch. We note that prior works differ in this setting but do not typically report what ratio is used. We see that using equal amounts of source and target data within each minibatch leads to the best performance. Notably, we also find that the inclusion of source-domain imagery is essential to see benefits from self-training—without any source imagery, $AP50_{FCS}$ drops from 64.5 to 59.3.

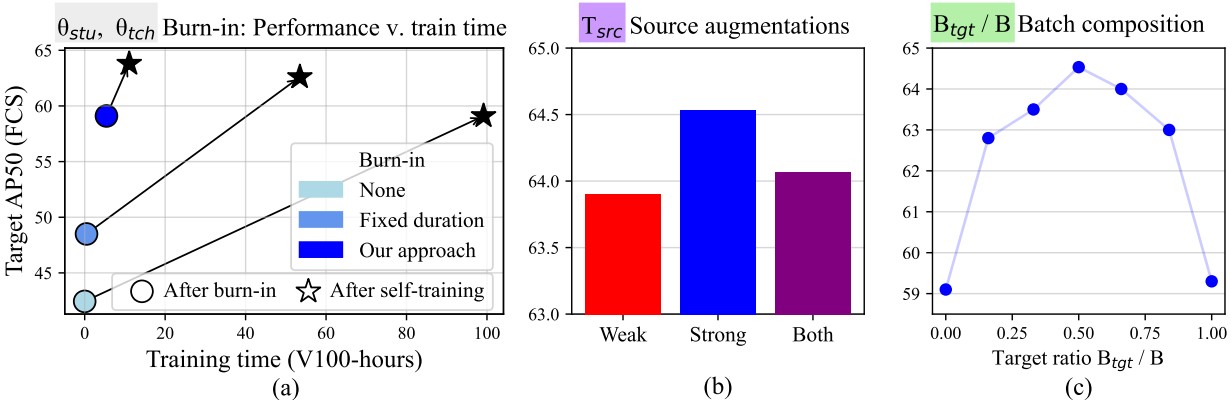

Figure 7: **Ablation studies.** **(a)** Our proposed burn-in strategy (Section 4) improves $AP50_{FCS}$ by +4.7 and reduces training time by 10x compared to no burn-in. **(b)** Strong source-data augmentations during self-training lead to better performance. **(c)** An equal ratio of source and target data during self-training leads to best performance.

$L_{distill}$ **Self-distillation.** In Table 1b we analyze the effects of our proposed multi-task soft distillation approach (see Section 4). We compare our approach with the "hard" pseudo-label approach used by prior work, where teacher predictions are post-processed with non-maximum suppression and a hard confidence threshold of 0.8 (Hoyer et al., 2023; Li et al., 2022b; Deng et al., 2021; Liu et al., 2021). For our proposed "soft" distillation method, we first sharpen teacher predictions at both detector stages using a sigmoid for objectness predictions and a softmax for classification predictions, both with a default temperature of 1. We see that our proposed soft targets improve performance compared to hard targets.

$L_{align}$ **Feature alignment.** Finally we investigate the use of feature alignment. We implement an adversarial feature alignment approach consisting of an image-level and instance-level feature discriminator (our implementation performs on par with SADA while being simpler to train; see Appendix A.3). In Table 1c, we show that feature alignment used in isolation (*i.e.* without self-training) offers performance gains up to 2.6 AP50. However, these performance gains are smaller than those seen from self-training (AP50$_{FCS}$ of 61.7 vs. 63.7, respectively). When used in combination with self-training techniques, the additional benefit of feature alignment drops to $\leq 0.2$ AP50$_{FCS}$. This suggests that self-training is currently the most promising avenue for progress and that more research is needed to develop complementary approaches. We also note that feature alignment approaches introduce training instability that may not be worth the small performance gain for practical use.

## 7 Discussion and Conclusions

In this work we proposed: the ALDI framework and an improved DAOD benchmarking methodology, providing a critical reset for the DAOD research community; a new dataset CFC-DAOD, increasing the diversity and real-world applicability of DAOD benchmarks; and a new method ALDI++ that advances the state-of-the-art. We conclude with key findings.

**Network initialization has an outsized impact.** We find that general advancements in computer vision eclipse progress in DAOD: a Resnet50-FPN source-only model outperforms all VGG-based DAOD methods, and a VitDet source-only model outperforms all Resnet50-FPN based DAOD methods. Similarly, simply adding stronger augmentations and EMA to source-only models leads to *better target-domain performance than some adaptation methods*, and including these upgrades during network initialization (burn-in) improves adaptation performance as well.

**DAOD techniques are helpful, but do not consistently achieve oracle-level performance as previously claimed (Li et al., 2022b; Chen et al., 2022; Cao et al., 2023).** Top-performing DAOD methods, including ALDI++, demonstrate improvements over source-only models (see Fig. 1 and Fig. 5). However, in contrast to previously-published results, no DAOD method consistently reaches oracle-level performance across datasets, architectures, and pre-training strategies, suggesting there is still room for improvement. The gap between DAOD methods and oracles is even larger for stronger architectures like VitDet. This is a promising area for future research.

**Benchmarks sourced from real-world domain adaptation challenges can help the community develop generally useful methods.** We find that DAOD methods do not necessarily perform equivalently across datasets (see Fig. 5). Diverse benchmarks are useful to make sure we are not overfitting to the challenges of one particular use case, while exposing and supporting progress in impactful applications. Our contributed codebase and benchmark dataset provide the necessary starting point to enable this effort.

**A lack of transparent comparisons has incentivized incremental progress in DAOD.** Most highly-performant prior works in DAOD are some combination of DANN (Ganin et al., 2016) and Mean Teacher (Tarvainen & Valpola, 2017) plus custom training techniques. Without fair comparisons it has been possible to propose near-duplicate methods that still achieve state-of-the-art performance due to hyperparameter tweaks. Our method ALDI++ establishes a strong point of comparison for *Align and Distill*-based approaches that will require algorithmic innovation to surpass.

**Validation is the elephant in the room.** All of our experiments, and all previously published work in DAOD, utilize a target-domain validation set to perform model and hyperparameter selection. This violates a key assumption in unsupervised domain adaptation: that no target-domain labels are available

to begin with. Prior work has shown that it may not be possible to achieve performance improvements in domain adaptation *at all* under realistic validation conditions (Musgrave et al., 2021; 2022; Kay et al., 2023). Therefore our results (as well as previously-published work) can really only be seen as an upper bound on DAOD performance. While this is valuable, further research is needed to develop effective unsupervised validation procedures for DAOD.

## Acknowledgments

This material is based upon work supported by: NSF CISE Graduate Fellowships Grant #2313998, MIT EECS department fellowship #4000184939, MIT J-WAFS seed grant #2040131, and Caltech Resnick Sustainability Institute Impact Grant "Continuous, accurate and cost-effective counting of migrating salmon for conservation and fishery management in the Pacific Northwest." Any opinions, findings, and conclusions or recommendations expressed in this material are those of the authors and do not necessarily reflect the views of NSF, MIT, J-WAFS, Caltech, or RSI. The authors acknowledge the MIT SuperCloud and Lincoln Laboratory Supercomputing Center for providing HPC resources Reuther et al. (2018). We also thank the Alaska Department of Fish and Game for their ongoing collaboration and for providing data, and Sam Heinrich, Neha Hulkund, Kai Van Brunt, Rangel Daroya, and Mark Hamilton for helpful feedback.

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

# A    Additional Experiments

## A.1    YOLO and DETR Architectures

To demonstrate the architecture-agnosticism of our framework and enable further research, we implement ALDI for the one-stage detection architecture YOLOv5 Redmon et al. (2016); Jocher et al. (2023) and the transformer-based architecture Deformable DETR Carion et al. (2020); Zhu et al. (2020). Implementation details are further described in Appendix B.

Table 2: **ALDI-YOLO results.**

(a) CS → FCS

| Method | AP50 |
|---|---|
| Source-only (YOLOv5m) | 58.8 |
| SSDA-YOLO Zhou et al. (2023) (YOLOv5l) | 55.9 |
| **ALDI-YOLO (ours, YOLOv5m)** | **62.5** |
| Oracle (YOLOv5m) | 66.3 |

(b) Sim10k → CS

| Method | AP50 |
|---|---|
| Source-only | 75.0 |
| ALDI-YOLO (ours) | 75.0 |
| Oracle | 88.0 |

(c) CFC-DAOD

| Method | AP50 |
|---|---|
| Source-only | 60.2 |
| ALDI-YOLO (ours) | 52.4 |
| Oracle | 76.7 |

Table 3: **DETR results on Cityscapes → Foggy Cityscapes.** We use 800px input size for consistency with prior work.

| Method | AP50 |
|---|---|
| Source-only | 44.5 |
| SFA Wang et al. (2021) | 41.3 |
| MTTrans Yu et al. (2022) | 43.4 |
| PM-DETR Jia et al. (2023) | 44.3 |
| **ALDI-DETR (ours)** | **44.8** |
| Oracle | 50.0 |

## A.2    ViT and ConvNeXt backbones

Table 4: Sim10k → Cityscapes

| Method | $AP50_{CS}$ |
|---|---|
| ViT-B baseline | 81.7 |
| ALDI++ + ViT-B | 81.8 |
| ViT-B oracle | 89.8 |

Table 5: CFC Kenai → Channel

| Method | $AP50_{Channel}$ |
|---|---|
| ViT-B baseline | 69.0 |
| ALDI++ + ViT-B | 71.1 |
| ViT-B oracle | 76.7 |

For completeness we show results using ALDI++ in combination with VitDet-B Li et al. (2022a) in Table 4 (Sim10k → Cityscapes) and Table 5 (CFC Kenai → Channel). We see that ALDI continues to demonstrate improvements over baselines even as overall architectures get stronger, though these improvements are smaller in magnitude than VitDet-B results on the CS → FCS dataset.

We also demonstrate the performance of ALDI++ with even larger backbones to examine how performance and domain gaps change. We show results from VitDet-L in Table 6, and with ConvNeXt-L in Table 7. Results are similar to our main results; we continue to see improvements of baselines, oracles, and ALDI++ in these settings.

## A.3    Adversarial Feature Alignment

We report additional ablations for the adversarial feature alignment network(s) used, comparing our implementations of image-level alignment and instance-level alignment with a baseline and SADA. As we see in

Table 6: Cityscapes → Foggy Cityscapes

| Method | AP50$_{FCS}$ |
|---|---|
| ViT-L baseline | 70.2 |
| ALDI++ + ViT-L | 76.1 |
| ViT-L oracle | 77.4 |

Table 7: Cityscapes → Foggy Cityscapes

| Method | AP50$_{FCS}$ |
|---|---|
| ConvNext-L baseline | 58.9 |
| ALDI++ + ConvNext-L | 63.3 |
| ConvNext-L oracle | 64.1 |

Table 8a, Table 8b, and Table 8c, the best settings to use differ by dataset. By default our feature alignment experiments in Sec. 6.1 of the main paper use both instance and image level alignment. See Appendix B.6 below for further implementation details.

Table 8: **Comparison of adversarial alignment methods. (a)** Cityscapes → Foggy Cityscapes. We see that our implementations outperform SADA Chen et al. (2021) while being simpler. Image-level alignment is best, followed by Image + Instance. **(b)** Sim10k → Cityscapes. Instance-level alignment is best. **(c)** CFC Kenai → Channel. Image + Instance is best. We see there is no consistently-best strategy across datasets; however, we note that for all datasets, the benefit of using adversarial feature alignment is smaller than self-training (see Sec. 6.3 of the main paper).

(a)

| Method | AP50$_{FCS}$ |
|---|---|
| Source-only | 51.9 |
| SADA | 54.2 |
| Image-level (ours) | 55.8 |
| Instance-level (ours) | 54.3 |
| Image + Instance (ours) | 54.9 |

(b)

| Method | AP50$_{CS}$ |
|---|---|
| Source-only | 70.8 |
| Image-level (ours) | 71.8 |
| Instance-level (ours) | 73.3 |
| Image + Instance (ours) | 71.5 |

(c)

| Method | AP50$_{Channel}$ |
|---|---|
| Source-only | 65.8 |
| Image-level (ours) | 65.2 |
| Instance-level (ours) | 66.0 |
| Image + Instance (ours) | 66.9 |

## A.4 Visualizing Alignment

We investigate the overlap of source and target data in the feature space of different methods. For each method, we pool the highest-level feature maps of the backbone, either globally ("image-level") or per instance ("instance-level"). We then embed the pooled feature vectors in 2D space using PCA for visual inspection (see Fig 8). We also compute a dissimilarity score based on FID Heusel et al. (2017), by fitting Gaussians to the source and target features and then computing the Fréchet distance between them.

## A.5 Teacher update

We compare other approaches to updating the teacher during self-training vs. using exponential moving average in Table 9. We see that EMA significantly outperforms using a fixed teacher (*i.e.* vanilla self-training, where pseudo-labels are generated once before training) as well as using the student as its own teacher without EMA.

Table 9: Comparison of teacher update approaches on Cityscapes → Foggy Cityscapes. Mean teacher greatly outperforms other options.

| Method | AP50$_{FCS}$ |
|---|---|
| Source-only baseline | 51.9 |
| No update (vanilla self-training) | 52.9 |
| Student is teacher | 53.8 |
| EMA (mean teacher) | 63.5 |

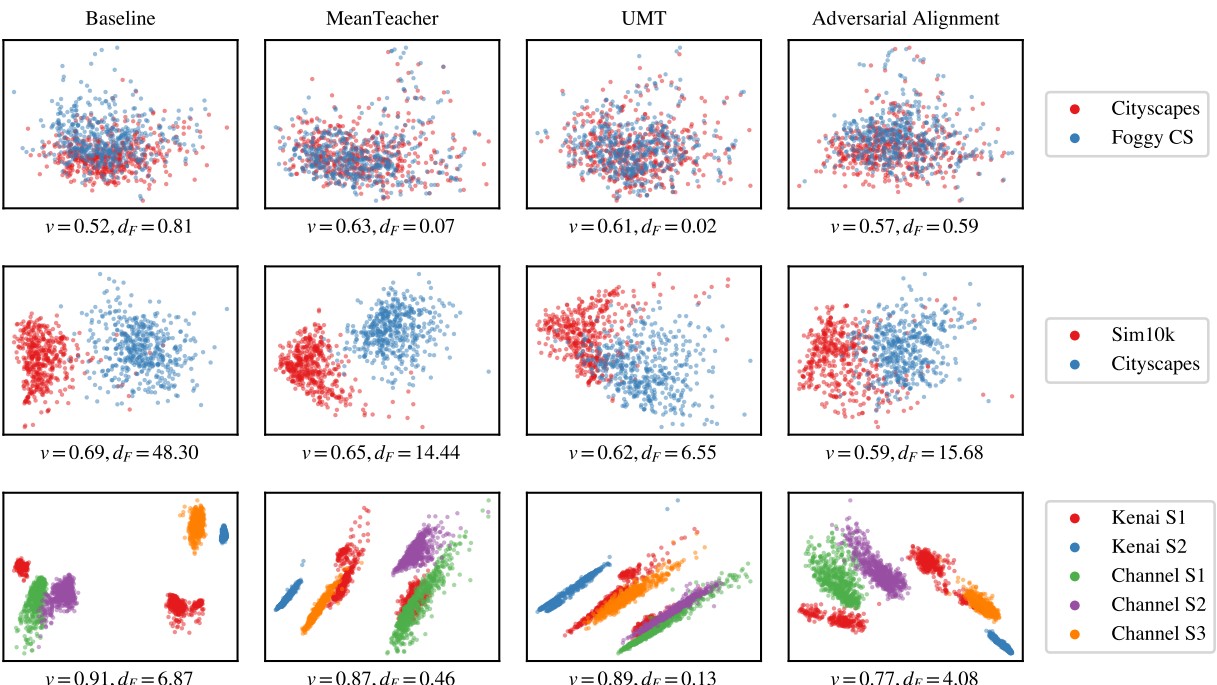

Figure 8: Embedding of pooled features from the final backbone layer in 2D space using PCA. We compare: (1) A source-only baseline, (2) The base settings from Section 6.3 ("MeanTeacher"), (3) UMT (which utilizes image-to-image translation), and (4) MeanTeacher + adversarial feature alignment using image-level features. The ratio of variance explained by the first two PCA components is given by $v$ and a dissimilarity score between source and target features is given by $d_F$. $d_F$ is lower than the baseline for all alignment methods and does roughly match the overall visual trend in feature overlap. In all cases, the simple MeanTeacher model significantly reduces the distance between source and target data even though there is no explicit alignment criterion, even resulting in a smaller $d_F$ than adversarial alignment methods for CS $\rightarrow$ FCS & CFC Kenai $\rightarrow$ Channel.

## A.6 Example of (Un)Fair Comparisons

In Fig. 9 we show a case study of why fair comparisons are impactful for DAOD research. We compare two similarly-performing prior works, AT and MIC, and see that implementation inconsistencies have led to nontransparent comparisons between the two methods. Notably, the originally reported results even used different ground truth test labels. When re-implemented on top of the same modern framework using ALDI, we are able to fairly compare the two methods for the first time.

## A.7 Method-specific Source-Only Models

Our protocol for training fair source-only models introduced in Section 6.1 is to utilize all techniques from the methods being studied that do not need target data to run. For simplicity, in the main paper we have only displayed the source-only model that utilizes the same components as ALDI++, though these settings differ slightly from the other methods studied. In Fig. 10 we show an alternate view in which for each method we train a bespoke source-only model using the exact same training settings as the DAOD method. The main difference is the set of image augmentations used, except for SADA, which also does not use EMA. We see that there is only a small variation in the strength of the source-only models corresponding to each method, so our choice to only visualize one in the main paper for simplicity is reasonable. The exception is SADA, whose source-only model is significantly weakened by not using EMA.

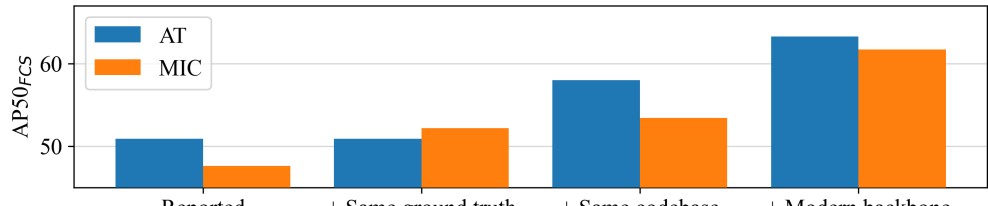

Figure 9: **Effects of fair and modernized comparison between MIC and AT.** Here we show an example of why fair and modern comparisons are necessary for making principled progress in DAOD. Moving left to right: **(1)** Published results report a difference of 3.3 AP50 on Cityscapes → Foggy Cityscapes between the two methods; **(2)** However the authors used *different truth test labels*, and when this is corrected we see that the originally-published MIC model actually outperforms the originally-published AT model; **(3)** The authors also used different object detection libraries (Detectron2 for AT and maskrcnn-benchmark for MIC); when we re-implement them on top of ALDI (still using the VGG-16 backbones proposed in the original papers), we see that AT significantly outperforms MIC, but **(4)** These performance differences are less pronounced when using a modern backbone, indicating that for practical use there is less difference between these two methods than previously reported.

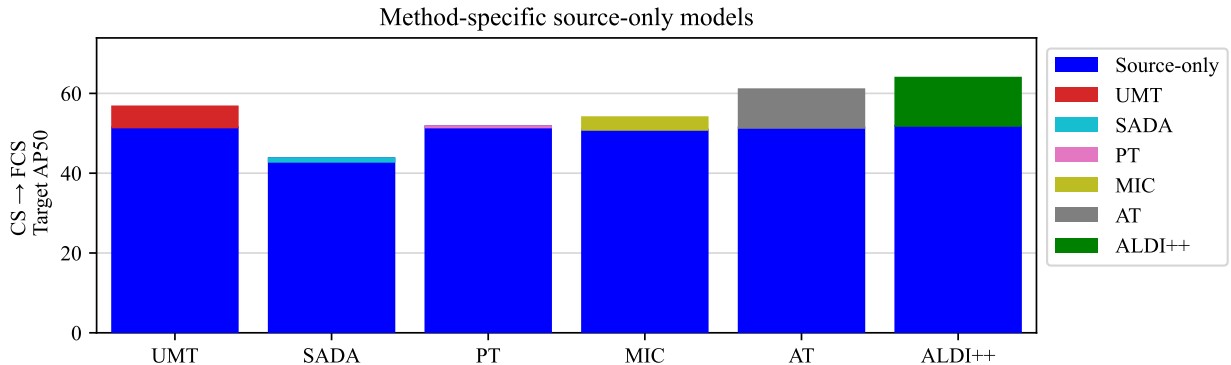

Figure 10: **Comparing methods to method-specific source-only models** that use the exact same training settings as the DAOD method in question. These results use ImageNet pre-training. Most of the bespoke source-only models perform very similarly, so in the main paper we only visualize one representative source-only model. The exception is SADA, whose corresponding source-only model performs worse due to the lack of EMA during training. See Appendix A.7.

# B    Implementation Details

## B.1    ALDI-YOLO

We use an Detectron2 implementation of YOLOv5m as our starting point. All hyperparameter settings are identical to those of ALDI++.

We implement soft distillation for YOLOv5 as follows. We compute an "objectness" (foreground/background) loss for each proposal, and compute classification and localization losses for pseudo-foreground labels only. Given pre-softmax student logits $l$ and teacher logits $\hat{l}$:

$$L_{obj,soft} = BCE(l_{obj}, \hat{l}_{obj}) \tag{14}$$

$$L_{cls,soft} = CE(l_{cls}, \alpha(\hat{l}_{cls}) \tag{15}$$

$$L_{loc,soft} = CIOU(l_{loc}, \alpha(\hat{l}_{loc})) \tag{16}$$

$$L_{distill,soft} = L_{obj,soft} + L_{cls,soft} + L_{loc,soft} \tag{17}$$

Where BCE is the binary cross-entropy loss, CE is the cross-entropy loss, CIOU is the Complete IoU loss **?**, and $\alpha$ is still a function of the post-softmax scores. See **??** for a visual depiction.

## B.2    ALDI-DETR

We use an Detectron2 implementation of Deformable DETR as our starting point. There is not an established technique for using soft knowledge distillation in end-to-end transformer-based queries like those of DETR; thus, we use hard distillation with a pseudo label threshold of 0.8. Similar to prior work Wang et al. (2021); Yu et al. (2022); Jia et al. (2023), we disable the EMA update for object query parameters.

Table 10: **Settings to reproduce five prior works and our method ALDI++. Burn-in:** fixed duration (Fixed), our approach (Ours, Section 4). **Augs. $T_{src}, T_{tgt}$:** Random flip (F), multi-scale (M), crop & pad (CP), color jitter (J), gaussian blur (B), cutout (DeVries & Taylor, 2017) (C), MIC (Hoyer et al., 2023). $\frac{1}{2}$: augs used on half the images in the batch. $\frac{B_{tgt}}{B}$: Target-domain portion of minibatch of size $B$. **Postprocess:** Processing of teacher preds before distillation: sigmoid/softmax (Sharpen), sum class preds for pseudo-objectness (Sum), conf. thresholding (Thresh), NMS. $L_{distill}$: Distillation losses: hard pseudo-labels (Hard), continuous targets (Soft). $L_{align}$: Feature alignment losses: image-level adversarial (Img), instance-level adversarial (Inst), image-to-image translation (Img2Img). †: settings used in ALDI implementation (last column) but not in the original implementation (second-to-last column). $^{at}$: source-only and oracle results sourced from Li et al. (2022b).

| Method | $\theta_{stu}, \theta_{tch}$ Burn-in | $T_{src}$ | $T_{tgt}$ | $\frac{B_{tgt}}{B}$ | Post-process | $L_{distill}$ | $L_{align}$ |
|---|---|---|---|---|---|---|---|
| Source-only | – | F, M†, C†, B†, E† | – | 0.0 | – | – | – |
| SADA (Chen et al., 2021) | – | F, M† | F, M† | 0.5 | – | – | Img, Inst |
| PT (Chen et al., 2022) | Fixed | F, M† | F, M†, J, C, B | 0.3 | Sharpen, Sum | Soft | – |
| UMT (Deng et al., 2021) | – | F†, M† | F†, M†, CP, J, B | 0.5 | Thresh, NMS | Hard | Img2Img |
| MIC (Hoyer et al., 2023) | – | F, M† | F, M†, J, B, MIC | 0.5 | Thresh, NMS | Hard | Img, Inst |
| AT (Li et al., 2022b) | Fixed | F, M†, J$^{\frac{1}{2}}$, C$^{\frac{1}{2}}$, B$^{\frac{1}{2}}$ | F, M†, J, C, B | 0.3 | Thresh, NMS | Hard | Img |
| **ALDI++** | Ours | F, M, J, C, B | F, M, J, B, MIC | 0.5 | Sharpen | Soft | – |
| Oracle | – | – | F, M†, J†, C†, B† | 1.0 | – | – | – |

## B.3    Re-implementations of Other Methods

Here we include additional details regarding our re-implementations of prior work on top of the ALDI framework. All hyperparameters are reported in Table 10. We visualize our implementations in Fig. 11.

### B.3.1 Adaptive Teacher (Li et al., 2022b)

Adaptive Teacher (AT) uses the default settings from the base configuration in Table 2 of the main paper, plus an image-level alignment network. For fair reproduction, we used the authors' alignment network implementation instead of our own for all AT experiments.

### B.3.2 MIC (Hoyer et al., 2023)

We reimplemented the masked image consistency augmentation as a Detectron2 `Transform` in our framework for efficiency. We also implemented MIC's "quality weight" loss re-weighting procedure, though in our experiments we found that it makes performance slightly worse (AP50 on Foggy Cityscapes of 62.8 vs. 63.1 without).

### B.3.3 Probabilistic Teacher (Chen et al., 2022)

Probabilistic Teacher (PT) utilizes: (1) a custom Faster R-CNN architecture that makes localization predictions probabilistic, called "Gaussian R-CNN", (2) a focal loss objective, (3) learnable anchors. We ported implementations of these three components to our framework. Note that we first had to burn in a Gaussian R-CNN, so PT was not able to use the exact same starting weights as other methods.

### B.3.4 SADA (Chen et al., 2021)

We port the official implementation of SADA to Detectron2. Note that SADA does not include burn-in or self-training, so the base implementation is the Detectron2 baseline config.

### B.3.5 Unbiased Mean Teacher (Deng et al., 2021)

Our implementation mirrors the $\text{UMT}_{SCA}$ configuration from Deng et al. (2021).

## B.4 Faster R-CNN Losses

Here we describe the standard Faster R-CNN losses before describing how we modify them into "soft" distillation losses. Faster R-CNN consists of two stages: a region proposal network and the region-of-interest heads.

### B.4.1 Region proposal network (RPN):

**Inputs.** The RPN takes as input:

1. Features extracted by a backbone network (*e.g.* a Resnet-50 with feature pyramid network in most of our experiments).

2. A set of anchor boxes that represent the initial candidates for detection.

**Outputs.** For each anchor, the RPN predicts two things:

1. A binary classification called "objectness" representing whether the content inside the anchor box is foreground or background.

2. Regression targets for the anchor, representing adjustments to the box to more closely enclose any foreground objects.

**Computing the loss.** In order to evaluate these predicted proposals, each proposal is matched to either foreground or background based on its intersection-over-union with the nearest ground truth box. Based

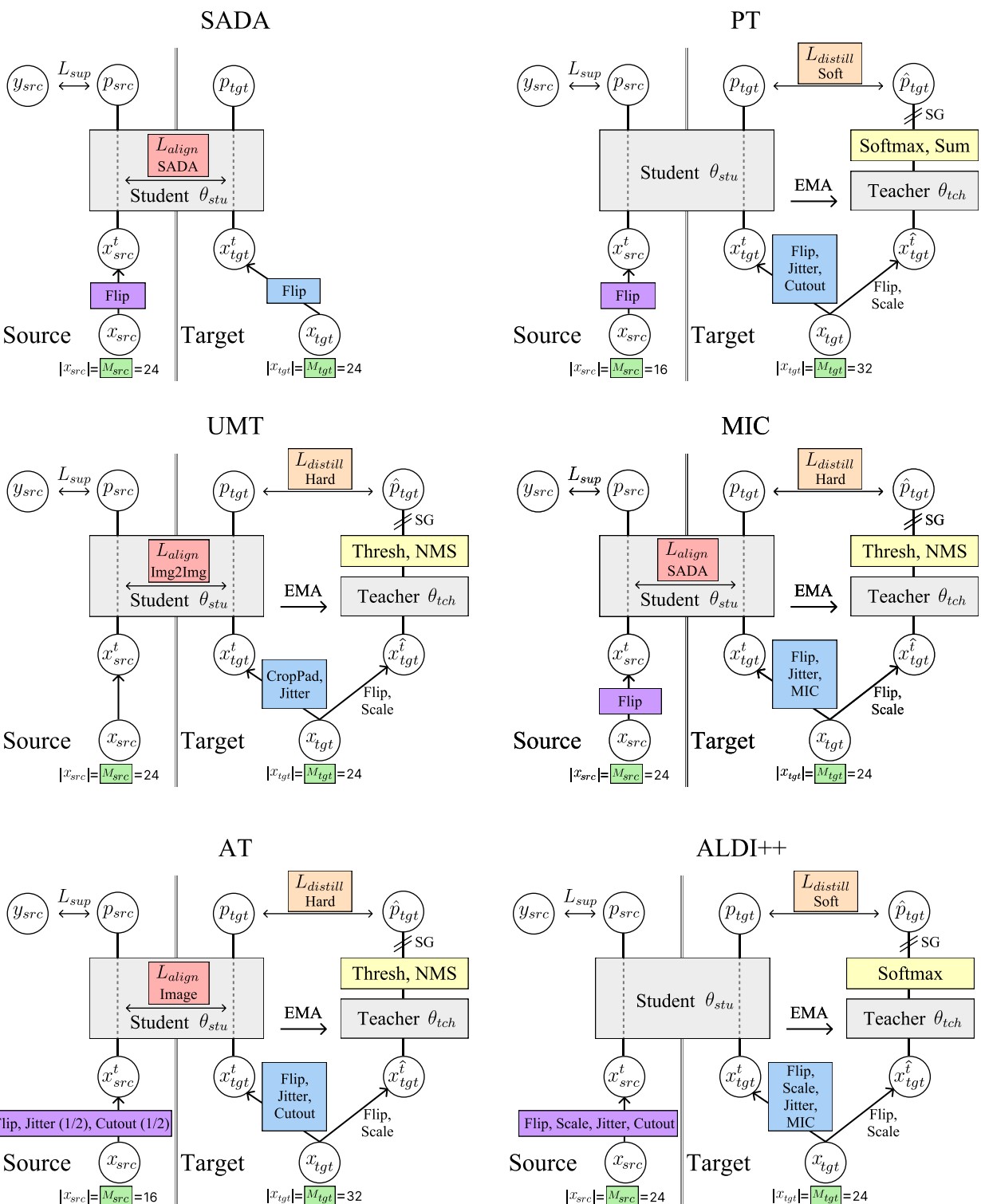

Figure 11: **Visual depiction of ALDI settings for reproducing prior work.**

on these matches, in the Detectron2 default settings a binary cross-entropy loss is computed for (1) and a smooth L1 loss is computed for (2).

A key challenge in Faster R-CNN is the severe imbalance between foreground and background anchors. To address this, a smaller number of proposals are sampled for computing the loss (256 in the default settings) with a specified foreground ratio (0.5 in the default setting). Objectness loss is computed for all proposals, while the box regression loss is computed only for foreground proposals (since it is undefined how the network should regress background proposals).

### B.4.2 Region of interest (ROI) heads:

**Inputs.** The ROI heads take as input:

1. Proposals from the RPN. In training, these are sampled at a desired foreground/background ratio, similar to the procedure used for computing the loss in the RPN. Note, however, that these will be different proposals than those used to compute RPN loss. In the Detectron2 defaults, 512 RPN proposals are sampled as inputs to the ROI heads at a foreground ratio of 0.25.

2. Cropped backbone features, extracted using a procedure such as ROIAlign (He et al., 2017). These are the features in the backbone feature map that are "inside" each proposal.

**Outputs.** The ROI heads then predict for each proposal:

1. A multi-class classification.

2. Regression targets for the final bounding box, representing adjustments to the box to more closely enclose any foreground objects.

**Computing the loss.** Predicted boxes are matched with ground truth boxes again based on intersection-over-union in order to compute the loss. By default we compute a cross-entropy loss for (1) and a smooth L1 loss for (2). (2) is again only computed for foreground predictions.

### B.5 Soft Distillation Losses for Faster R-CNN

Distillation losses are computed between teacher predictions and student predictions. One option is select the teacher's most confident predictions based on a confidence threshold parameter to be "pseudo-labels." These take the place of ground truth boxes in the standard Faster R-CNN losses for the student. We refer to this approach as using "hard targets."

In contrast, here we describe how we compute "soft" losses using intermediate outputs from the teacher to guide the student without thresholding.

**RPN.** The teacher and student RPNs start with the same anchors. We use the same sampling procedure described in B.4.1 for choosing proposals for loss computation. Importantly, we ensure the *same* proposals are sampled from the teacher and student so that they can be directly compared. We postprocess the teacher's objectness predictions with a sigmoid function to sharpen them. We then compute a binary cross-entropy loss between the teacher's post-sigmoid outputs and student's objectness predictions. We also compute a smooth L1 loss between the teacher's RPN regression predictions and the student's RPN regression predictions. Regression losses are only computed on proposals where the teacher's post-sigmoid objectness score is $\geq 0.8$.

**ROI heads.** The second stage of Faster R-CNN predicts a classification and regression for each RPN proposal; therefore, we need the input proposals to the student and teacher to be the same in order to directly compare their outputs. To achieve this, during soft distillation we initialize the student and teacher's ROI heads with the student's RPN proposals—intuitively, we want the teacher to tell the student "what to do with" its proposals from the first stage.

We postprocess the teacher's classification predictions with a softmax to sharpen them, then compute a cross-entropy loss between the teacher's post-softmax predictions and the student's classification predictions. We also compute a smooth L1 loss between the teacher's regression predictions and the student's regression predictions. We only compute regression losses where the teacher's top-scoring class prediction is not the background class.

### B.6 Adversarial Feature Alignment

We implement two networks to perform adversarial alignment at the image level and instance (bounding box) level. Our approach is inspired by Faster R-CNN in the Wild (Chen et al., 2018) and SADA (Chen et al., 2021).

**Image-level alignment.** We build an adversarial discriminator network that takes in backbone features at the image level. By default we use the "p2" layer of the feature pyramid network as described in Lin et al. (2017). We use a simple convolutional head consisting of one hidden layer. Our defaults result in this `torch` module:

```
ConvDiscriminator(
    (model): Sequential(
      (0): Conv2d(256, 256,
                  kernel_size=(3, 3),
                  stride=(1, 1))
      (1): ReLU()
      (2): AdaptiveAvgPool2d(output_size=1)
      (3): Flatten(start_dim=1, end_dim=-1)
      (4): Linear(in_features=256,
                  out_features=1,
                  bias=True)
    )
  )
```

**Instance-level alignment.** We also implement an instance-level adversarial alignment network that takes as input the penultimate layer of the ROI heads classification head. By default, our instance level discriminator consists of one hidden fully-connected layer. Our defaults result in this `torch` module:

```
FCDiscriminator(
    (model): Sequential(
      (0): Flatten(start_dim=1, end_dim=-1)
      (1): Linear(in_features=1024,
                  out_features=1024,
                  bias=True)
      (2): ReLU()
      (3): Linear(in_features=1024,
                  out_features=1,
                  bias=True)
    )
)
```

## C   Experiment Details

### C.1   Backbone Pretraining

In our experiments, we evaluate two different backbones: a ResNet-50 (He et al., 2016) with Feature Pyramid Network (Lin et al., 2017), and a ViT-B (Dosovitskiy et al., 2020) with ViTDet (Li et al., 2022a). Both backbones are pre-trained on the ImageNet-1K classification and the COCO instance segmentation (Lin et al., 2014) tasks. In addition, the ViT-B backbone is also pre-trained using the masked autoencoder objective proposed in He et al. (2022).

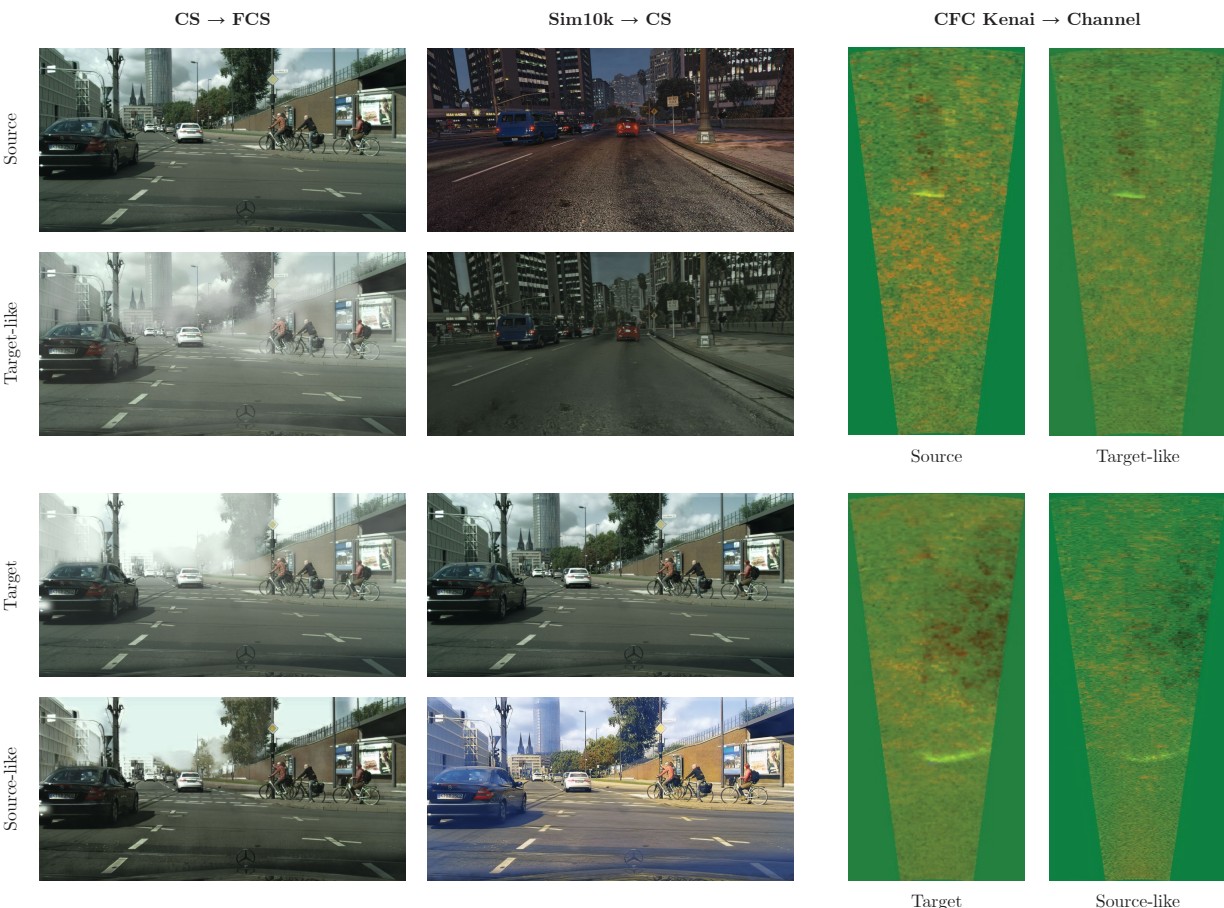

Figure 12: **Exemplary results of our CycleGAN models.** Source and target are the original images. Target-like and source-like are images translated by CycleGAN. Since FCS is derived from CS, CS → FCS is the only case in which we have paired images and can therefore show the translation from source into target-like and from target into source-like for the same example.

## C.2 Image-to-Image Translation

In contrast to the adversarial alignment in feature space as in SADA (Chen et al., 2021), UMT (Deng et al., 2021) aligns the domains in image (*i.e.* pixel) space. This is achieved by training and using an unpaired image-to-image translation model to try to transform images from the source dataset into images that look like images from the target dataset ("target-like") and vice-versa ("source-like"). We follow Deng et al. (2021) by using the CycleGAN (Zhu et al., 2017) image-to-image translation model. We train the CycleGAN for 200 epochs (Cityscapes ↔ Foggy Cityscapes, Sim10k ↔ CS) or 20 epochs (Kenai ↔ Channel) and respectively select the best model according to the average Fréchet inception distance (FID) (Heusel et al., 2017) between the source & source-like and the target & target-like images in the training dataset. For FID computation, we use the clean-fid implementation proposed in Parmar et al. (2022). We compute FID on the training datasets as UMT only uses translated images thereof, which is why we are only interested in the best fit on the training data. We follow Deng et al. (2021) by then generating source-like and target-like dataset using the selected model ahead of time, before the training of the main domain adaptation method. We note that tuning CycleGAN's hyperparameters or using other image-to-image translation methods could possibly improve UMT's performance however for the fair reproduction we use the defaults. We show some exemplary results of our CycleGAN models that are used to train UMT (Deng et al., 2021) in Fig 12.

### C.3 Other Training Settings

We fix the total effective batch size at 48 samples for fair comparison across all experiments. For training, we perform each experiments on 8 Nvidia V100 (32GB) GPUs distributed over four nodes. We use the MIT Supercloud (Reuther et al., 2018).

## D  CFC-DAOD Dataset Details

Like other DAOD benchmarks, CFC-DAOD consists of data from two domains, source and target.

### D.1 Source data

**Train:** In CFC-DAOD, the source-domain training set consists of training data from the original CFC data release, i.e., video frames from the "Kenai left bank" location. We have used the 3-channel "Baseline++" format introduced in the original CFC paper (Kay et al., 2022). For experiments in the ALDI paper, we subsampled empty frames to be around 10% of the total data, resulting in 76,619 training images. For reproducibility, we release the exact subsampled set. When publishing results on CFC-DAOD, however, researchers are allowed to use the orignial CFC training set however they see fit and are not required to use our subsampled "Baseline++" data.

**Validation:** The CFC-DAOD Kenai (source) validation set is the same as the original CFC validation set. We use the 3-channel "Baseline++" format from the original CFC paper. There are 30,454 validation images.

### D.2 Target data

**Train:** In CFC-DAOD, the target-domain "training" set consists of new data from the "Kenai Channel" location in CFC. These frames should be treated as unlabeled for DAOD methods, but labeled for Oracle methods. We also use the "Baseline++" format, and use the authors' original code for generating the image files from the original video files for consistency. There are 29,089 target train images.

**Test:** The CFC-DAOD target-domain test set is the same as the "Kenai Channel" test set from CFC. We use the "Baseline++" format. There are 13,091 target test images. Researchers should publish final mAP@Iou=0.5 numbers on this data, and may use this data for model selection for fair comparison with prior methods.

## E  The ALDI Codebase

We release ALDI as an open-source codebase built on a modern detector implementation. The codebase is optimized for speed, accuracy, and extensibility, training up to 5x faster than existing DAOD codebases while requiring up to 13x fewer lines of code. These qualities make our framework valuable for practitioners developing detection models in real applications, as well as for researchers pushing the state-of-the-art in DAOD.

### E.1 Detection Framework

We designed the ALDI codebase to be lightweight and extensible. For this reason, we build on top of a recent version of Detectron2 (Wu et al., 2019). The last tagged release of Detectron2 was `v0.6` in November 2021, however there have been a number upgrades since then leading to state-of-the-art performance. Thus, we use a fixed version that we call `v0.7ish` based off of an unofficial pull request for `v0.7`, commit `7755101` dated August 30 2023. We include this version of Detectron2 as a `pip`-installable submodule in the ALDI codebase for now, noting that once the official version is released it will no longer need to be a submodule (*i.e.* it will be able to be directly installed through `pip` without cloning any code).

Table 11: **Open-source codebases in domain adaptive object detection.** Existing methods use a variety of different detector implementations, including deprecated frameworks (maskrcnn-benchmark) and versions (Detectron2 < v0.6). In contrast, ALDI is built on top of a modern framework, optimized for training speed, and is able to reproduce all five of these methods while requiring fewer lines of code (LOC) than any individual existing implementation. Our codebase can serve as a strong starting point for future research.

| Codebase | Faster R-CNN Implementation | Backbone | Input size | LOC |
|---|---|---|---|---|
| UMT (Deng et al., 2021) | faster-rcnn.pytorch | VGG-16 | 600px | 19k |
| SADA (Chen et al., 2021) | maskrcnn-benchmark | Resnet50-FPN | 800px | 7k |
| PT (Chen et al., 2022) | Detectron2 v0.5 | VGG-16 | 600px | 3.4k |
| MIC (Hoyer et al., 2023) | maskrcnn-benchmark | Resnet50-FPN | 800px | 20k |
| AT (Li et al., 2022b) | Detectron2 v0.3 | VGG-16 | 600px | 4k |
| ALDI (Ours) | Detectron2 ∼v0.7 | Resnet50-FPN | 1024px | 1.5k |

Our codebase makes no modifications to the underlying Detectron2 code, making it a lightweight standalone framework. This is in contrast to existing DAOD codebases (see Table 11) that often duplicate and modify the underlying framework as part of their implementation. By building *on top of* Detectron2 rather than *within* it, our codebase is up to 13x smaller than other DAOD codebases while providing more functionality. We note that in Table 11, other codebases implement a single method while ours supports all methods studied.

### E.2 Speedups

We found significant bottlenecks in training in other Detectron2-based codebases. Notably, we found that dataloaders and transform implementations were inefficient. These included, for instance:

- Converting tensors back and forth between torch, numpy, and PIL during augmentation. We addressed this, reimplementing transforms as needed so that everything stays in torch.

- Using the random hue transform from torchvision. We found minimal changes in performance from disabling this component of the ColorJitter transform.

- Using separate dataloaders for weakly and strongly augmented imagery. We instead use a single dataloader per domain, with a hook to retrieve weakly augmented imagery before strong augmentations are performed.

We reimplemented the dataloaders and augmentation strategies used by AT, MIC, and others to be more efficient, leading to a 5x speedup in training time per image compared to AT.

## F  Tabular Results

## G  Qualitative Results

We visualize predictions from all models on Cityscapes → Foggy Cityscapes in Fig. 13, Sim10k → Cityscapes in Fig. 14, and CFC-DAOD in Fig. 15.

**Cityscapes → Foggy Cityscapes** We choose 1 random frame from each city in the validation set (Munster, Lindau, and Frankfurt) and show the 0.005 and 0.02 fog levels. We see that, compared to ALDI++, UMT and PT consistently suffer from more false positive detections, while other methods display both false positive and false negative detections. AT and MIC perform qualitatively similary to ALDI++ in many cases. AT suffers from more false positives than ALDI++ in all locations but fewer false negatives in Lindau. MIC

Table 12: **Results with ResNet50-FPN backbones and ImageNet pre-training.** Previously-published results are shown in gray. Best results for each benchmark are in **bold** and second-best are underlined.

| Method | CS → Foggy CS | Sim10k → CS | CFC Kenai → Channel |
|---|---|---|---|
| Source-only | (23.5) 51.9 | (35.5) 61.4 | 67.3 |
| UMT (Deng et al., 2021) | (41.7) 56.7 | (43.1) 41.8 | 64.9 |
| SADA (Chen et al., 2021) | (44.0) 42.9 | 55.7 | 61.6 |
| PT (Chen et al., 2022) | (47.1) 51.5 | (55.1) 64.3 | 67.0 |
| MIC (Hoyer et al., 2023) | (47.6) 54.0 | 64.3 | 71.9 |
| AT (Li et al., 2022b) | (50.9) 61.0 | 56.7 | 69.2 |
| **ALDI++ (Ours)** | **63.9** | **69.1** | **72.6** |
| Oracle | (42.7) 62.4 | (66.4) 83.3 | 77.1 |

Table 13: **Results with ResNet50-FPN backbones and COCO pre-training.** Best results for each benchmark are in **bold** and second-best are underlined.

| Method | CS → Foggy CS | Sim10k → CS | CFC Kenai → Channel |
|---|---|---|---|
| Source-only | 59.1 | 76.8 | 66.7 |
| UMT (Deng et al., 2021) | 61.4 | 58.7 | 61.2 |
| SADA (Chen et al., 2021) | 54.2 | 71.8 | 58.9 |
| PT (Chen et al., 2022) | 59.2 | 70.6 | 69.0 |
| MIC (Hoyer et al., 2023) | 61.7 | 73.1 | 75.5 |
| AT (Li et al., 2022b) | 63.3 | 72.0 | 69.1 |
| **ALDI++ (Ours)** | **66.8** | **77.8** | **76.1** |
| Oracle | 67.2 | 86.1 | 73.8 |

Table 14: **Results with VitDet-B backbones and ImageNet pre-training.** Best results are in **bold** and second-best are underlined.

| Method | CS → Foggy CS |
|---|---|
| Source-only | 68.1 |
| UMT (Deng et al., 2021) | 63.9 |
| SADA (Chen et al., 2021) | 54.4 |
| PT (Chen et al., 2022) | 65.8 |
| MIC (Hoyer et al., 2023) | 70.6 |
| AT (Li et al., 2022b) | 66.6 |
| **ALDI++ (Ours)** | **71.0** |
| Oracle | 74.5 |

performs better than ALDI++ in this randomly-selected Lindau frame. In the Frankfurt example, we see one downside of the Foggy Cityscapes benchmark: since detection annotations are generated programatically from segmentations, strange false positives can exist in the ground truth. Our new dataset CFC-DAOD addresses this problem by focusing directly on object detection.

**Sim10k → Cityscapes** In Fig. 14 we compare results of all methods on the Sim10k → Cityscapes benchmark on two random images from each location in the Cityscapes validation set. In Munster, we see that all methods, including ALDI++, struggle with differentiating overlapping cars in the first image. In the second

image, ALDI and MIC outperform other methods in differentiating cars in groups with less overlap than the first example. In Lindau, most prior work exhibits significantly more false negatives than ALDI++ and AT. AT, however, merges detections of multiple cars, leading to false negatives as well. In Frankfurt, most methods again exhibit significant false negatives, while ALDI++ and MIC show small false positive detections. Across all locations, MIC and SADA predict large false positives caused by foreground street lines.

**CFC-DAOD** In Fig. 15 we compare results of all methods on the CFC-DAOD Kenai → Channel benchmark. We select two random images from each camera view in the test set. These camera views represent different range windows of the sonar camera, with Stratum1 the nearest-range (leading to the highest resolution and largest fish), followed by Stratum2, and finally Stratum3 is the longest-range (lowest resolution and smallest fish). We can see that each Stratum exhibits its own challenges in differentiating fish from the background, dealing with low signal-to-noise ratios, and differences in target size. In Stratum1, all methods except SADA detect the easily identifiable large fish in the first frame, but all methods suffer from the same false negative in the second frame. ALDI++ and UMT predict the same false positive near the edge of the field of view. In Stratum2, all methods suffer from false positives caused by background texture, and UMT, SADA, and PT suffer false negatives. In Stratum3, all methods fail to detect the furthest-range fish which are highly occluded by background texture, but exhibit fewer false positives than the other strata.

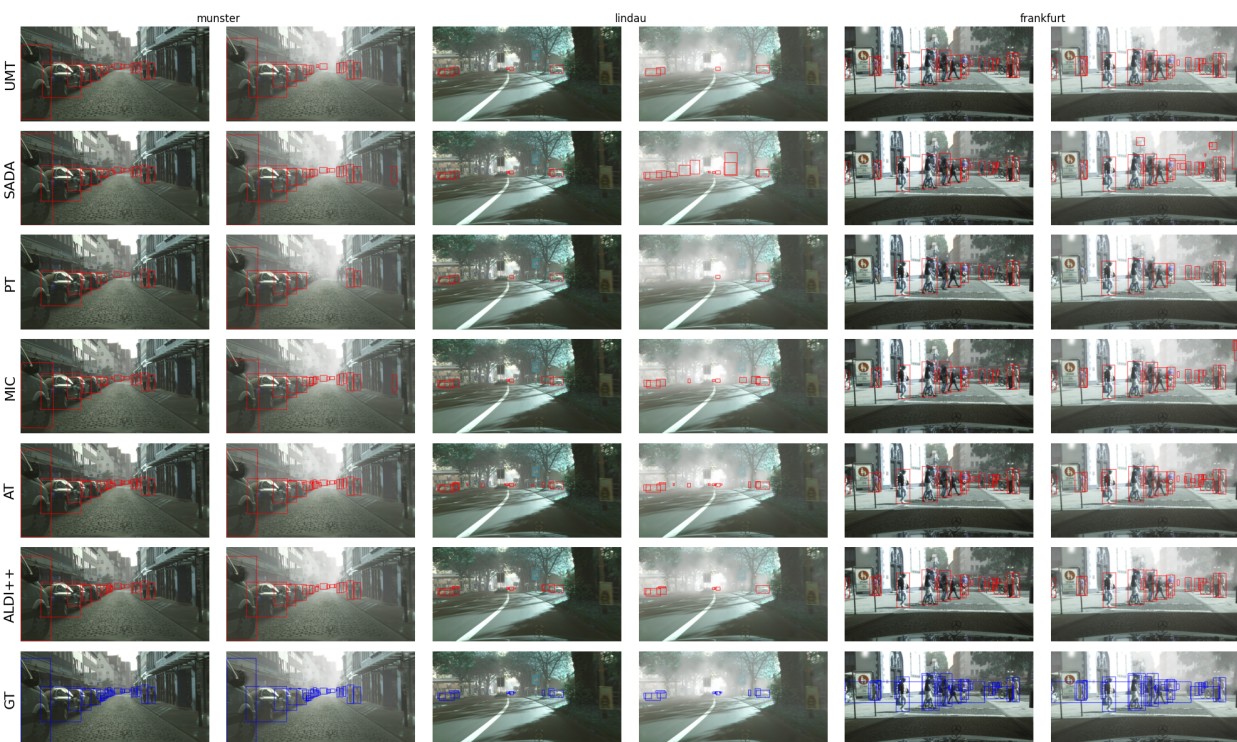

Figure 13: Qualitative results on Foggy Cityscapes. Best viewed maginified.

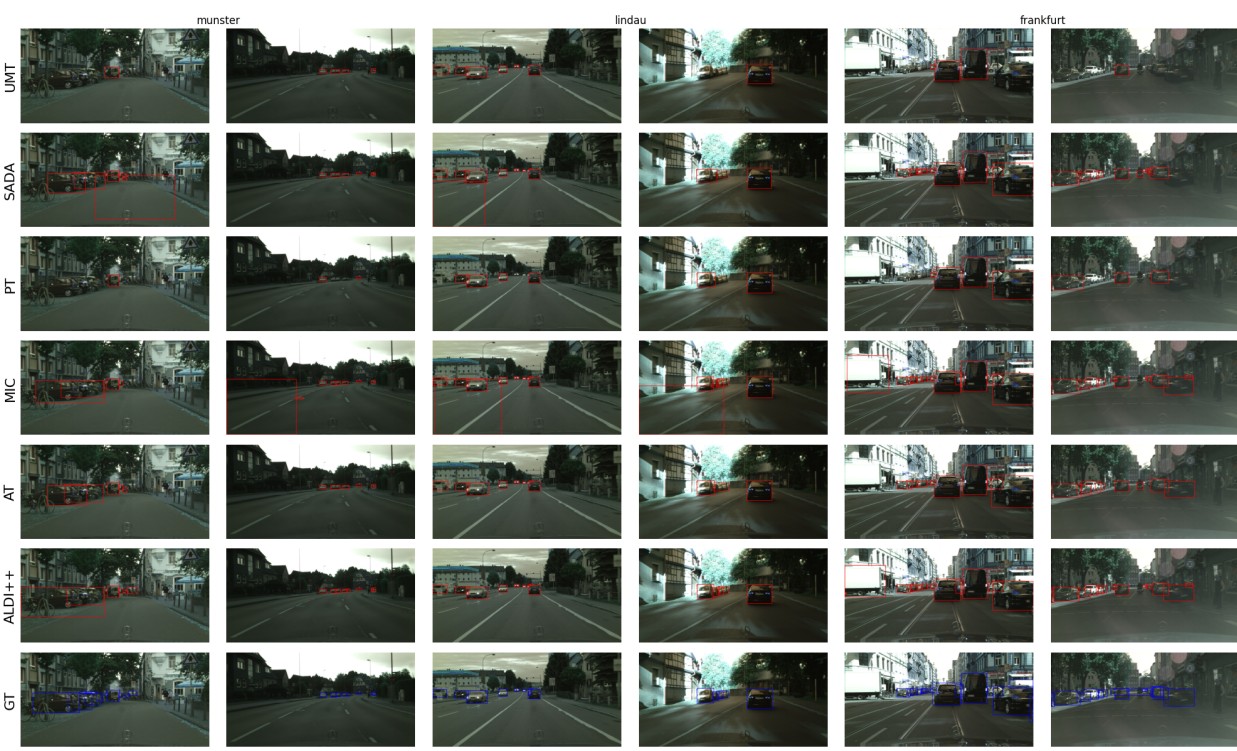

Figure 14: Qualitative results on Sim10k → Cityscapes. Best viewed maginified.

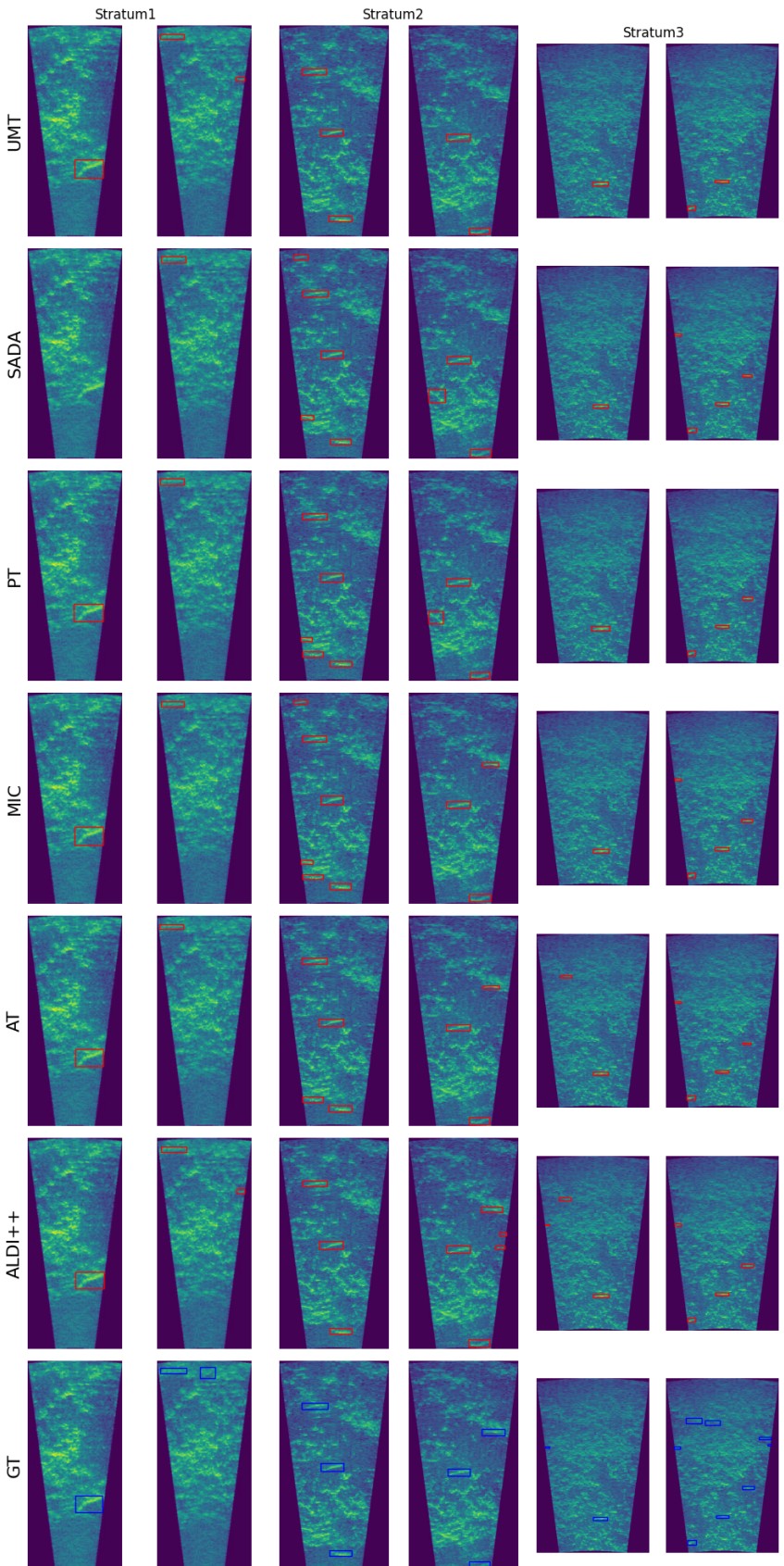

Figure 15: Qualitative results on the CFC-DAOD test set. Best viewed maginified.

