# OpenReview forum: "Align and Distill: Unifying and Improving Domain Adaptive Object Detection"
_TMLR — Accepted by TMLR_

### Review · Reviewer_6uSg · 2024-09-14

**Summary Of Contributions:**

This paper introduces Align and Distill (ALDI), a framework that unifies common components of domain adaptive object detection (DAOD) methods to address past benchmarking flaws, ensuring a fair comparison between methods. Specifically,  ALDI compares five DAOD methods that use self-training and feature alignment. The authors also introduce ALDI++, a new method that achieves state-of-the-art results by including three improvements: (i) a strategy to improve out-of-distribution generalization before self-training, (ii) the use of soft distillation losses instead of hard ones, and (iii)  a revised training approach with strong regularization on both target and source data, and a balanced number of source and target samples in each minibatch.

The authors show that some evaluated DAOD methods overestimate their improvement compared to their source-only model. This source-only model is specifically designed to exclude performance gains unrelated to DAOD methods by removing only the components that involve target data. To further investigate the generalizability of DAOD methods, the authors also introduce an extended version of the Caltech Fish Counting Dataset for the DAOD task.

**Audience:**

Yes

**Broader Impact Concerns:**

No concern about the ethical implications of the work.

**Claims And Evidence:**

Yes

**Requested Changes:**

The following proposed adjustments could further strengthen an already solid work:
1. The authors could include a discussion on the limitations of the proposed ALDI framework, particularly regarding which methods can be re-implemented.
2. It would be helpful to clarify whether the validation set used to tune the design choices is FCS as for the ablation study. Additionally, the authors could report the main hyperparameters of the five methods re-implemented in the ALDI framework compared to their original implementation.
3. The authors should clarify the categories of methods that can be compared to the source-only baseline proposed in the paper.

**Strengths And Weaknesses:**

Strengths
1. ALDI creates a fair benchmarking framework for domain adaptive object detection (DAOD) methods, helping to understand whether performance improvements are due to domain adaptation techniques or other factors, an ambiguity that characterizes DAOD methods.
2. The authors re-implement 5 methods for DAOD, which proves the possibility of using this framework for future work.
3. The authors introduce ALDI++, a new method that achieves state-of-the-art performance on three benchmarks. They also introduce a new dataset for evaluating the generalizability of DAOD methods.
4. The authors' claims are supported by extensive experiments.

Weaknesses
1. While implementing DAOD methods within a single framework like Detectron2 has its advantages, I wonder how difficult it would be to use the ALDI framework with an object detection model not supported by Detectron2, such as the one-stage object detector YOLO.
2. The authors acknowledge using a target-domain validation set for model and hyperparameter selection, but it is unclear which dataset was used for this purpose. Additionally, it is unclear the advantage of tuning ALDI++ on this dataset, especially compared to the re-implemented methods, which were not tuned, even though some design choices (such as input resolution and batch size) were modified compared to the original implementation.
3. It is unclear why the ablation study does not use the final model. For example, the default setting used in the ablation study involves hard thresholding, rather than soft thresholding, which is one of the advantages introduced by ALDI++.
4. In Figure 10, the authors compare DAOD methods to their source-only model. They also mention that SADA does not use EMA during training. Given this, is it fair to compare methods that do not use EMA with the source-only baseline used in the paper?

---

> ### Author Response · Authors · 2025-01-07
> **Response to Reviewer 6uSg and plan to address requested revisions**
>
> Dear Reviewer 6uSg,
>
> Thank you very much for your careful reading and review of our manuscript. We are encouraged to hear that the key takeaways we presented were clear: namely that our framework helps “understand whether performance improvements are due to domain adaptation techniques or other factors, an ambiguity that characterizes DAOD methods”, and that our contributed method “ALDI++, [...] achieves state-of-the-art performance on three benchmarks” and these findings are “supported by extensive experiments”.
>
> We also thank you for the constructive feedback you have provided. We will address your noted weaknesses and requested changes here as well as in an updated version of the manuscript.

---

> > ### Author Response · Authors · 2025-01-07
> > **Requested revision 1**
> >
> > **Weakness 1.** *While implementing DAOD methods within a single framework like Detectron2 has its advantages, I wonder how difficult it would be to use the ALDI framework with an object detection model not supported by Detectron2, such as the one-stage object detector YOLO.*
> > **Requested change 1.** *The authors could include a discussion on the limitations of the proposed ALDI framework, particularly regarding which methods can be re-implemented.*
> >
> > Thank you for pointing out that “implementing DAOD methods within a single framework like Detectron2 has its advantages” -- we agree! First, please note that current state-of-the-art DAOD methods are predominantly built on top of the Faster R-CNN architecture. While there is indeed prior work that utilizes YOLO [1] architectures for DAOD, we note crucially that they are not state-of-the-art and currently under-perform Faster R-CNN based approaches. However, as you point out, even though most DAOD methods utilize Faster R-CNN, algorithms differ in their choice of framework, making fair comparison difficult. Building on top of a state-of-the-art framework like Detectron2 allows us to address this, while also enabling a more extensible code design due to its modularity.
> >
> > Our framework, ALDI, is architecture-agnostic by design, and indeed this makes it possible to implement ALDI on top of YOLO as well. To address your request, we have done so, and have completed experiments for our ALDI-YOLO model on all datasets in our paper. The results are shown below, and we will include these as well as additional implementation details in a revised version of the manuscript.
> >
> > ALDI-YOLO performs on par with Faster R-CNN based methods on the Cityscapes to Foggy Cityscapes benchmark. ALDI-YOLO also achieves a new state-of-the-art for one-stage object detection, outperforming SSDA-YOLO [1] even with a smaller detector size (we use YOLOv5m while SSDA-YOLO uses YOLOv5l).
> >
> > Interestingly, the other benchmarks prove more challenging, supporting your observation that our “new dataset for evaluating the generalizability of DAOD methods” is a strength. This is an interesting area for future research to build upon our work. Thank you very much for the suggestion, and we hope our implementation of ALDI-YOLO will encourage further progress in this area!
> >
> > **Cityscapes**
> >
> > | Method | AP50 |
> > | ---- | ---- |
> > | Source only (ours) | 58.8 |
> > | SSDA-YOLO | 55.9 |
> > | **ALDI-YOLO** | **62.5** |
> > | Oracle | 66.3 |
> >
> > **Sim10k**
> >
> > | Method | AP50 |
> > | ---- | ---- |
> > | Source only (ours) | 75.0 |
> > | ALDI-YOLO | 75.0 |
> > | Oracle | 88.0 |
> >
> > **CFC-DAOD**
> >
> > | Method | AP50 |
> > | ---- | ---- |
> > | Source only (ours) | 60.2 |
> > | ALDI-YOLO | 52.4 |
> > | Oracle | 76.7 |
> >
> > [1] Zhou, Huayi, Fei Jiang, and Hongtao Lu. "SSDA-YOLO: Semi-supervised domain adaptive YOLO for cross-domain object detection." Computer Vision and Image Understanding 229 (2023): 103649.

---

> ### Author Response · Authors · 2025-01-07
> **Requested revisions 2–3**
>
> **Weakness 2.** *The authors acknowledge using a target-domain validation set for model and hyperparameter selection, but it is unclear which dataset was used for this purpose. Additionally, it is unclear the advantage of tuning ALDI++ on this dataset, especially compared to the re-implemented methods, which were not tuned, even though some design choices (such as input resolution and batch size) were modified compared to the original implementation.*
> **Weakness 3.** *It is unclear why the ablation study does not use the final model. For example, the default setting used in the ablation study involves hard thresholding, rather than soft thresholding, which is one of the advantages introduced by ALDI++.*
> **Requested revision 2.** *It would be helpful to clarify whether the validation set used to tune the design choices is FCS as for the ablation study. Additionally, the authors could report the main hyperparameters of the five methods re-implemented in the ALDI framework compared to their original implementation.*
>
> Thank you for this question, we agree that additional clarity would be helpful here! The two points in question—hyperparameter tuning, and our ablation study—are closely related, so we will answer them together.
>
> We designed our ablation study to investigate the new methodological techniques we introduce (robust burn-in and soft distillation) as well as to be a general investigation into align-and-distill approaches in DAOD (encompassing common techniques such as augmentation strategy and batch composition). We note that in many cases, these components are not identified as key design decisions nor ablated in prior work—for example, the top-performing prior works, AT [1] and MIC [2], do not mention or ablate the batch composition or source-domain augmentation strategies, yet we find that these choices account for much of the performance difference between them (more examples of these discrepancies can be seen in Figure 9 of the appendix). Our ablation study emphasizes the importance of these often-overlooked design decisions. Thank you for asking for clarification around this, and we will add this further clarification around the experimental design to the manuscript.
>
> The ablation study was conducted, as you note, on the CS -> FCS benchmark, so in this regard our method is tuned to FCS. However we did not tune ALDI++ directly to the other datasets (see Section 6.2, where we note: “All methods (including ALDI++) use the same settings for all benchmarks”) yet still achieve state-of-the-art results. We believe this demonstrates the generality of our findings, as ALDI++ seems to generalize well across datasets without additional tuning.
>
> Thank you for recommending additional clarification around the hyperparameters used by prior work. We provide these details in Table 6 of the appendix. Wherever possible, we used the exact hyperparameters used by authors of the original implementation.
>
> [1] Li, Yu-Jhe, et al. "Cross-domain adaptive teacher for object detection." Proceedings of the IEEE/CVF Conference on Computer Vision and Pattern Recognition. 2022.
>
> [2] Hoyer, Lukas, et al. "MIC: Masked image consistency for context-enhanced domain adaptation." Proceedings of the IEEE/CVF conference on computer vision and pattern recognition. 2023.
>
> **Weakness 4.** *In Figure 10, the authors compare DAOD methods to their source-only model. They also mention that SADA does not use EMA during training. Given this, is it fair to compare methods that do not use EMA with the source-only baseline used in the paper?*
> **Requested revision 3.** *The authors should clarify the categories of methods that can be compared to the source-only baseline proposed in the paper.*
>
> This is an astute observation! Thank you for the close read. Indeed, it is not quite fair to compare SADA to the same source-only and oracle models as the other methods, because SADA is the only method that does not use EMA. According to our proposed benchmarking protocol, it would indeed be appropriate to compare each method to a representative source-only model that uses the exact same components as the DAOD model. We chose to display a single source-only and oracle model in Figure 1 for clarity, since most methods have nearly identical components, however we included a more detailed comparison with bespoke, fair source-only models in Figure 10 of the appendix. As expected, SADA’s source-only model performs worse overall due to its lack of EMA, so the relative under-performance of SADA is lessened. We agree that this is not clear in the main text, and we will add an additional point of clarification to a revised version of the manuscript to direct readers to the comparisons of Figure 10.

---

> > ### Author Response · Authors · 2025-01-24
> > **Follow up regarding revisions**
> >
> > Dear Reviewer 6uSg,
> >
> > Thank you very much again for your valuable feedback.
> >
> > As noted above, we have addressed your requested revisions. You can find them in the revised manuscript in the following locations:
> >
> > **Requested revision 1:** We have implemented ALDI on top of the YOLO (and Deformable DETR) architectures, and included experimental results in Appendix G.
> >
> > **Requested revision 2:** Please see response above for clarification.
> >
> > **Requested revision 3:** Please see response above for clarification. We have added a reference to the additional experiments in Section 6.1 of the main text.
> >
> > Thank you very much and we look forward to any additional feedback or questions you may have.

---

### Review · Reviewer_vcow · 2024-09-20

**Summary Of Contributions:**

This paper identifies key benchmarking issues in Domain Adaptive Object Detection, such as outdated architectures, inconsistent training, and limited benchmark diversity, which have led to inflated performance claims and incremental progress. To address this, this paper introduces modern baseline architectures, re-implements training and evaluation protocols, and establishes a new benchmark dataset. With fair comparisons, this paper shows prior methods may be instead sub-optimal, highlighting the field's slow advancement.   Finally, this paper proposes two general techniques that significantly outperform previous works.

**Audience:**

Yes

**Broader Impact Concerns:**

There's no ethical implications of the work

**Claims And Evidence:**

Yes

**Requested Changes:**

1. Authors are encouraged to conduct experiments over modern "detection" framework, such as DINO[1]
2. Authors should define $T_{strong}$ properly.
3. Authors are encouraged to conduct experiments over state-of-the-art visual backbones, such as ConvNext-L[2] or CLIP/ViT-L[3]
4. Authors are encouraged to explain why ALDI++ outperforms oracle model when using R50-FPN (ImageNet pre-train)


---

Reference:

[1] DINO: DETR with Improved DeNoising Anchor Boxes for End-to-End Object Detection.  Zhang et al.
[2] A ConvNet for the 2020s.  Liu et al.
[3] OpenCLIP.

**Strengths And Weaknesses:**

Strengths:
1. This paper is well-written, clearly identifying the issue and its underlying causes.
2. This paper effectively clarifies the inconsistencies in training and evaluation protocols used in prior works. By isolating adaptation-specific techniques to create fairer baselines, it clearly exposes the incremental progress of previous methods.  I believe such results help the community to re-think the commonly adopted methodologies and review criteria in this field.
3. The new dataset proposed by this paper is indeed novel and enhances benchmarking diversity.
4. This paper demonstrates that two simple techniques outperform all previous methods. The results are convincing, especially when compared against modernized baselines.

---

Weaknesses

1. This paper claims to modernize the baseline architectures (P1), however, the detection framework (Faster-RCNN) is in fact obsolete, contradicting to its claim.
2. $T_{strong}$ in Section 4.1 is not defined properly
3. In Figure 6, it seems that stronger baseline architectures (ViTDet-B vs. R50-FPN) have less domain gap.  As this paper has shown that prior works may be sub-optimal, it's unclear if the proposed ALDI++ helps when using using state-of-the-art baseline architectures, such as ConvNext-L or CLIP/ViT-L.  One might conjecture that there would be very little or even no domain gap.
4. This paper does not explain why ALDI++ outperforms oracle when using R50-FPN (ImageNet pre-train).

---

> ### Author Response · Authors · 2025-01-07
> **Response to Reviewer vcow and plan to address requested revisions**
>
> Dear Reviewer vcow,
>
> Thank you very much for your detailed review and feedback on our paper. We are pleased to hear that the main takeaways were clear, and that our work “effectively clarifies the inconsistencies in training and evaluation protocols used in prior works”, “help[ing] the community to re-think the commonly adopted methodologies and review criteria in this field”. Indeed we hope our work will be impactful in orienting the field toward meaningful progress. We also thank you for noting that “the new dataset proposed by this paper is indeed novel and enhances benchmarking diversity” and “results are convincing, especially when compared against modernized baselines”. We hope that our dataset and method will also help the field.
>
> We also thank you for your constructive feedback, and will address your concerns individually here as well as with new experiments and revised text in an updated version of the manuscript.

---

> > ### Author Response · Authors · 2025-01-07
> > **Requested revision 1**
> >
> > **Weakness 1:** *“This paper claims to modernize the baseline architectures (P1), however, the detection framework (Faster-RCNN) is in fact obsolete, contradicting to its claim.”*
> > **Requested change 1:** *“Authors are encouraged to conduct experiments over modern "detection" framework, such as DINO”*
> >
> > While this is a logical concern, methods built on top of Faster R-CNN continue to be state-of-the-art in the field of domain adaptive object detection (DAOD). The methods we benchmarked (AT, MIC, PT, etc.) represent the current state-of-the-art and are all built on top of Faster R-CNN. There is indeed prior work that utilizes YOLO [1] and DETR [2] architectures for DAOD, however we note crucially that they are not state-of-the-art and currently under-perform Faster R-CNN based approaches. This motivated our choice of Faster R-CNN for the experiments and we hope you will agree this was a logical and justified choice.
> >
> > We emphasize that there are two components to a “modernized” object detector: the detection architecture (Faster R-CNN, YOLO, DETR, etc.) and the backbone (VGG, ResNet, ViT, etc). One key observation from our paper was that existing state-of-the-art DAOD methods were consistent in their architecture choice (Faster R-CNN) but inconsistent in their backbone choice (VGG vs. ResNet), leading to questions around whether the comparisons and rankings were fair. By equivalently modernizing the backbone choice—i.e. performing comparisons between methods where each method uses a ResNet or ViT backbone—we are able to get a more realistic picture of how these methods perform.
> >
> > That being said, we value your feedback and agree that there is value in further exploration of the effects of even more modern architectures and backbones within the context of DAOD. We are working to address your concerns and follow your recommendations for revisions as follows:
> >
> > 1. We will add clarification about the difference between detection “architectures” and “backbones” in the related work in order to make our claim about modernizing baselines more clear. We will post a revision with these changes in the coming days.
> >
> > 2. We have implemented a version of ALDI on top of the Deformable DETR architecture to address your request. Deformable DETR is currently the most advanced transformer-based architecture used in prior DAOD work, e.g. [2] and [3]. An advantage of the ALDI framework is that it is essentially architecture-agnostic: all that is required to adapt the framework to new architectures is an “align” module (which can be easily implemented with standard neural network feature representations) and a “distill” module (which can be as simple as hard pseudo-labeling). While we did not initially implement ALDI for DETR architectures for the reasons described above (namely, the prevalence of Faster R-CNN DAOD methods in the literature), DAOD methods for DETR currently utilize very similar alignment and distillation objectives, and adding this functionality into our framework can help encourage further progress in transformer-based methods as well—thank you for this suggestion! We are currently performing experiments with our ALDI-DETR implementation, and plan to post a revision to the paper with results in the coming days.
> >
> > [1] Zhou, Huayi, Fei Jiang, and Hongtao Lu. "SSDA-YOLO: Semi-supervised domain adaptive YOLO for cross-domain object detection." Computer Vision and Image Understanding 229 (2023): 103649.
> >
> > [2] Yu, Jinze, et al. "MTTrans: Cross-domain object detection with mean teacher transformer." European Conference on Computer Vision. Cham: Springer Nature Switzerland, 2022.
> >
> > [3] Jia, Peidong, et al. "PM-DETR: Domain adaptive prompt memory for object detection with transformers." arXiv preprint arXiv:2307.00313 (2023).

---

> > ### Author Response · Authors · 2025-01-07
> > **Requested revisions 2-4**
> >
> > **Weakness 2:** *“Tstrong in Section 4.1 is not defined properly”*
> > **Requested change 2.** *“Authors should define Tstrong properly.”*
> >
> > Thank you for pointing this out! This was a typo. We will post a revision to the paper in which Tstrong is defined properly.
> >
> > **Weakness 3:** *“In Figure 6, it seems that stronger baseline architectures (ViTDet-B vs. R50-FPN) have less domain gap. As this paper has shown that prior works may be sub-optimal, it's unclear if the proposed ALDI++ helps when using using state-of-the-art baseline architectures, such as ConvNext-L or CLIP/ViT-L. One might conjecture that there would be very little or even no domain gap.”*
> > **Requested change 3:** *“Authors are encouraged to conduct experiments over state-of-the-art visual backbones, such as ConvNext-L or CLIP/ViT-L”*
> >
> > This is indeed an interesting observation, and we agree that it is an interesting question for the field: to what extent will DAOD continue to be useful as baseline architectures and backbones improve? This question was not the main focus of our paper, and an in-depth exploration would be more appropriate for future work, however we agree that results with state-of-the-art visual backbones would be interesting to include as both a preliminary investigation of this question as well as a demonstration of the flexibility of our framework.
> >
> > To address your request, we have implemented ConvNeXt-L and CLIP-ViT-L backbones within the ALDI framework. We are currently running experiments with these backbones and plan to post a revision in the coming days as the experiments complete. Thank you for this feedback and for the suggestion!
> >
> > **Weakness 4:** *“This paper does not explain why ALDI++ outperforms oracle when using R50-FPN (ImageNet pre-train).”*
> > **Requested change 4:** *“Authors are encouraged to explain why ALDI++ outperforms oracle model when using R50-FPN (ImageNet pre-train)”*
> >
> > Thank you for this observation—we also found this result surprising! While we don’t have a definitive answer, there are a number of hypotheses for why this could occur. First, there is some noise in the Cityscapes dataset, since the bounding boxes are generated automatically from segmentation annotations. This is standard practice for Cityscapes and consistent with prior work, though it is another reason why we feel our new benchmark CFC-DAOD helps improve the landscape of DAOD benchmarking by focusing directly on detection. Second, there is still some amount of domain shift between the unlabeled “training” dataset and validation dataset of Foggy Cityscapes because they are sourced from different cities. It could be that, in some cases, self-training strategies can result in models that are more robust to additional distribution shifts than purely supervised methods. We believe this is an interesting avenue for future work that investigates align-and-distill based approaches in the context of domain generalization rather than adaptation.
> >
> > We will continue to investigate this result, and add our findings as well as an explanation to a revision of the manuscript within the next week. Thank you for the suggestion!

---

> > > ### Author Response · Authors · 2025-01-24
> > > **Follow up regarding revisions**
> > >
> > > Dear Reviewer vcow,
> > >
> > > Thank you very much again for your valuable feedback.
> > >
> > > As noted above, we have addressed your requested revisions. You can find them in the revised manuscript in the following locations:
> > >
> > > **Requested revision 1:** We have (1) Added additional clarifications on architectures vs. backbones in the Related Work and modified our use of the words "architecture" and "backbone" throughout, (2) Implemented ALDI on top of the Deformable DETR architecture, with experimental results in Appendix A.
> > >
> > > **Requested revision 2:** Fixed in Section 4.1 and 4.2.
> > >
> > > **Requested revision 3:** We have added ConvNeXt-L and ViT-L experiments, with results in Appendix G.
> > >
> > > **Requested revision 4:** Explanation added to Section 6.2.
> > >
> > > Thank you very much and we look forward to any additional feedback or questions you may have.

---

### Review · Reviewer_kcu1 · 2025-01-02

**Summary Of Contributions:**

This paper presents a unified benchmarking and implementation framework—Align and Distill (ALDI), to address the systemic pitfalls in domain adaptive object detection (DAOD). The authors identify that past results often overestimate performance due to underpowered baselines, inconsistent implementations, and reliance on outdated backbones. In response, they propose a modern DAOD evaluation protocol and introduce a new benchmark dataset (CFC-DAOD) that better reflects real-world diversity. They also present a new method, ALDI++, which achieves state-of-the-art results on multiple benchmarks and underscores the importance of careful model initialization and training pipelines. Despite these advances, the authors find that no DAOD method consistently reaches “oracle-level” performance across all architectures and pre-training strategies, revealing substantial room for improvement. They conclude that real-world benchmarks, transparent comparisons, and reliable validation procedures are essential for further progress, offering their framework, dataset, and approach as a critical reset for DAOD research.

**Audience:**

Yes

**Broader Impact Concerns:**

I do not observe obvious ethical concerns.

**Claims And Evidence:**

Yes

**Requested Changes:**

**Below is a list of suggested revisions**:
1. Provide additional justification or examples to demonstrate how the new fish detection setting materially broadens the scope of DAOD benchmarks.
2. While Faster R-CNN is a standard choice, consider adding preliminary results or discussions on more modern backbones (e.g., DETR or ViT-based) to solidify the claim that your method generalizes well beyond a traditional architecture.
3. Comprehensive visualizations showing detection results in both typical and challenging scenarios.
4. Failure case analysis, presenting visualizations of common failure cases.
5. Other weaknesses mentioned in the section above.

Addressing these points would significantly improve the paper’s clarity and contribution.

**Strengths And Weaknesses:**

### **Strengths**:
**[Clear Motivation and Problem Statement]** The paper provides a straightforward and well-motivated problem statement. The figures illustrate the system architecture, helping readers quickly grasp the overall workflow, and the authors highlight known issues in DAOD benchmarks that have long been recognized in the community.

**[Comprehensive Ablation Study]** In Section 6.3, the paper provides a wide-ranging ablation study for the ALDI++ model, examining factors such as source and target augmentation, batch composition ratio, multi-task soft distillation, and feature alignment. This thorough investigation reveals valuable insights into the framework’s design and is comprehensive.

**[Unification of Multiple DAOD Methods]** The ALDI framework successfully integrates multiple DAOD approaches into a single pipeline for fair, streamlined benchmarking. Its core design space, including supervised training, target-domain distillation, and adversarial alignment components, is technically reasonable.

**[Improved Baseline Protocol (P1)]** The authors devote significant attention to constructing a fair baseline model by comparing source-only and target-supervised methods. This step is often overlooked yet critical, as huge improvements can be observed from inadequate baselines.

---
### **Major Weaknesses:**

**[Weak Claims Regarding Benchmark Diversification P3(1)]** Authors have mentioned in P3 that DAOD benchmarks have focused on urban driving scenarios, thereby giving rise to an assumption that methods generalize across different domains and architectures. While it is true that the field has largely relied on driving benchmarks (due to their availability and standardization), it is not necessarily the case that prior work assumes generalizable applicability. Furthermore, although the authors highlight this gap, they only introduce one additional dataset component -- the fish detection setting -- which may not be sufficient to fully address the stated need for broader domain coverage.

**[Weak Claims on Outdated Backbones P3(2)]** Authors have claimed in P3 that reliance on older architectures hamper the perceived generality of DAOD methods. However, their own approach still uses Faster R-CNN, which is considered a somewhat dated backbone per se. Although a number of other methods have explored more advanced architectures (e.g., DETR, ViT-based detectors), the manuscript does not provide a concrete plan to move beyond the standard practice. As a result, the argument that most methods use outdated backbones remains unconvincing without a proposed alternative.

**[Insufficient Visualization]** The paper doesn't include any major big figure showing the domain adaptation object detection resuls. More comprehensive visual results, possibly in supplementary material would provide clearer evidence of the system’s capabilities. Without additional visual examples, it is difficult to assess the method’s performance, generalization, and consistency in more diverse scenarios.

---
### **Minor Weaknesses**
**[Modality-Specific Scope]** The title and introduction refer broadly to "Domain Adaptive Object Detection," but the paper deals exclusively with 2D detection using camera images. Given the importance of 3D detection in many modern driving applications, the authors should explicitly state that this work focuses on image-based 2D object detection.

**[Broken URLs in Abstract]** Two anonymous URLs provided in the abstract were no longer accessible at the time of review.

**[Missing Failure Case Analysis]** Although the paper describes strong results, it lacks a dedicated section on failure cases. Presenting examples of challenging or problematic scenarios, along with insights into the method’s shortcomings, would have provided a more complete picture of the model's capability.

---

> ### Author Response · Authors · 2025-01-07
> **Response to Reviewer kcu1 and plan to address requested revisions**
>
> Dear Reviewer kcu1,
>
> Thank you very much for the in-depth review of our manuscript and clear, detailed feedback. We are pleased to hear that you found our work to be “well-motivated”, “comprehensive”, “technically reasonable”, and noted that our findings highlight “overlooked yet critical” aspects of DAOD benchmarking.
>
> We also thank you for your constructive and clear feedback. We will address your noted weaknesses and requested changes here as well as in an updated version of the manuscript.

---

> > ### Author Response · Authors · 2025-01-07
> > **Requested revision 1**
> >
> > **Weakness 1.** *Claims Regarding Benchmark Diversification [...] it is not necessarily the case that prior work assumes generalizable applicability [...] they only introduce one additional dataset component -- the fish detection setting -- which may not be sufficient to fully address the stated need for broader domain coverage.* **Requested change 1.** *Provide additional justification or examples to demonstrate how the new fish detection setting materially broadens the scope of DAOD benchmarks.*
> >
> > Thank you for this suggestion, we agree that the manuscript would benefit from additional examples of the value our dataset provides to DAOD benchmarking. Note, however, we do not claim that our dataset “fully address[es] the stated need for broader domain coverage" -- this would be far too much to claim. However, we argue that it certainly increases the diversity of DAOD benchmarks. The value of this can be seen most clearly in Figure 5 and Section 6.2, where we find that the ranking of methods differs across datasets, thus focusing on a single dataset such as Cityscapes is not guaranteed to result in methods that generalize across the numerous real-world applications of domain adaptive object detection. Our dataset provides an additional point of comparison for researchers to consider when developing methods.
> >
> > Still, we value your feedback, and agree that additional justification or examples could help further demonstrate the value of our contribution. With this in mind, we are working to add the following to address your concerns:
> >
> > 1. We will remove the language in P3 that states “The underlying assumption is that methods will perform equivalently across application domains and backbone architectures”, since we agree with you that prior work does not explicitly make this claim. Thank you for this suggestion.
> >
> > 2. In accordance with your other requested revisions 3 and 4 (which we further respond to below), we will provide additional visualizations and failure case analysis of methods on both our benchmark and the others, calling attention to new insights that can be gleaned from the fish detection setting.

---

> > ### Author Response · Authors · 2025-01-07
> > **Requested revision 2**
> >
> > **Weakness 2.** *Claims on Outdated Backbones P3(2)*
> > **Requested change 2.** *Consider adding preliminary results or discussions on more modern backbones (e.g., DETR or ViT-based) to solidify the claim that your method generalizes well beyond a traditional architecture.*
> >
> > While this is a logical concern, methods built on top of Faster R-CNN continue to be state-of-the-art in the field of domain adaptive object detection (DAOD). The methods we benchmarked (AT, MIC, PT, etc.) represent the current state-of-the-art and are all built on top of Faster R-CNN. There is indeed prior work that utilizes DETR [1, 2] architectures for DAOD, however we note crucially that they are not state-of-the-art and currently under-perform Faster R-CNN based approaches. This motivated our choice of Faster R-CNN for the experiments and we hope you will agree this was a logical and justified choice.
> >
> > We emphasize that there are two components to a “modern” object detector: the detection architecture (Faster R-CNN, YOLO, DETR, etc.) and the backbone (VGG, ResNet, ViT, etc). One key observation from our paper was that existing state-of-the-art DAOD methods were consistent in their architecture choice (Faster R-CNN) but inconsistent in their backbone choice (VGG vs. ResNet), leading to questions around whether the comparisons and rankings were fair. By equivalently modernizing the backbone choice—i.e. performing comparisons between methods where each method uses a ResNet or ViT backbone—we are able to get a more realistic picture of how these methods perform. Note that, with respect to your requested change, we have already benchmarked our method ALDI++ as well as other prior work with ViT backbones; see Figures 1 and 6 and Section 6.3 of the main paper and Appendix A.3 for these ViT results.
> >
> > That being said, we value your feedback and agree that there is value in further exploration of the effects of even more modern architectures and backbones within the context of DAOD. We are working to address your concerns and follow your recommendations for revisions as follows:
> >
> > 1. We will add clarification about the difference between detection “architectures” and “backbones” in the related work in order to make our claim about modernizing baselines more clear.  We will post a revision with these changes in the coming days.
> >
> > 2. We have implemented a version of ALDI on top of the Deformable DETR architecture to address your request. Deformable DETR is currently the most advanced transformer-based architecture used in prior DAOD work, e.g. [1] and [2]. An advantage of the ALDI framework is that it is essentially architecture-agnostic: all that is required to adapt the framework to new architectures is an “align” module (which can be easily implemented with standard neural network feature representations) and a “distill” module (which can be as simple as hard pseudo-labeling). While we did not initially implement ALDI for DETR architectures for the reasons described above (namely, the prevalence of Faster R-CNN DAOD methods in the literature), DAOD methods for DETR currently utilize very similar alignment and distillation objectives, and adding this functionality into our framework can help encourage further progress in transformer-based methods as well—thank you for this suggestion! We are currently performing experiments with our ALDI-DETR implementation, and plan to post a revision to the paper with results in the coming days.
> >
> > 3. We have implemented an additional ViT backbone, CLIP-ViT-L, which is larger than the ViT-B we have already benchmarked and utilizes stronger CLIP pre-training. We are currently running experiments with this backbone and will also include these experiments in a revision in the coming days.
> >
> > [1] Yu, Jinze, et al. "MTTrans: Cross-domain object detection with mean teacher transformer." European Conference on Computer Vision. Cham: Springer Nature Switzerland, 2022.
> >
> > [2] Jia, Peidong, et al. "PM-DETR: Domain adaptive prompt memory for object detection with transformers." arXiv preprint arXiv:2307.00313 (2023).

---

> > ### Author Response · Authors · 2025-01-07
> > **Requested revisions 3–4**
> >
> > **Weakness 3.** *Insufficient Visualization*
> > **Requested change 3.** *Comprehensive visualizations showing detection results in both typical and challenging scenarios.*
> > **Minor weakness 3.** *Missing Failure Case Analysis*
> > **Requested change 4.** *Failure case analysis, presenting visualizations of common failure cases.*
> >
> > Thank you very much for these recommendations. We agree that this would improve the manuscript, and are working to add more qualitative comparisons and failure case analysis to a revision that will be posted in the next week.
> >
> > **Minor weakness 1.** *Modality-Specific Scope*
> >
> > Thank you for bringing this to our attention. We have reviewed related work and confirmed that the term “domain adaptive object detection” is widely used to refer to 2D object detection, while “domain adaptive 3D object detection” is typically used for work dealing with 3D object detection. Still, we agree that being explicit in this will add clarity, and we will add a clarifying sentence to our manuscript to address your concern.
> >
> > **Minor weakness 2.** *Broken links*
> >
> > Thank you for pointing this out. We have fixed the broken links.

---

> > > ### Author Response · Authors · 2025-01-24
> > > **Follow up regarding revisions**
> > >
> > > Dear Reviewer kcu1,
> > >
> > > Thank you very much again for your valuable feedback.
> > >
> > > As noted above, we have addressed your requested revisions. You can find them in the revised manuscript in the following locations:
> > >
> > > **Requested revision 1:** We have (1) Changed the language in P3 on page 2, and (2) Included additional visualizations and performance/failure case analysis in Appendix G.
> > >
> > > **Requested revision 2:** We have (1) Added additional clarifications on architectures vs. backbones in the Related Work and modified our use of the words "architecture" and "backbone" throughout, (2) Implemented ALDI on top of both the Deformable DETR architecture and Vit-L backbones, with experimental results in Appendix A.
> > >
> > > **Requested revisions 3 and 4:** Again please see the additional visualizations and performance/failure case analysis in Appendix G.
> > >
> > > **Minor weaknesses:** Fixed as described above.
> > >
> > > Thank you very much and we look forward to any additional feedback or questions you may have.

---

### Author Response · Authors · 2025-01-16
**All requested revisions addressed**

Dear Reviewers and Action Editor,

We have uploaded a revision to our manuscript that addresses all requested revisions of all reviewers. For clarity, new content is colored in blue and removed content is colored in red with strikethroughs.

We thank the reviewers for their clear and constructive feedback. We are pleased that the key takeaways and significance of our work was clear to all reviewers. In particular, reviewers brought attention to the following strengths our four key contributions:

**1. ALDI, a unified benchmarking framework for DAOD.** As Reviewer 6uSg notes, “ALDI creates a fair benchmarking framework for domain adaptive object detection (DAOD) methods, helping to understand whether performance improvements are due to domain adaptation techniques or other factors, an ambiguity that characterizes DAOD methods.” This was indeed the main goal of ALDI: to unify common approaches such that we can rigorously study and improve them. The generality of the framework was noted by Reviewer kcu1: “The ALDI framework successfully integrates multiple DAOD approaches into a single pipeline for fair, streamlined benchmarking. Its core design space, including supervised training, target-domain distillation, and adversarial alignment components, is technically reasonable” and this benefit was also noted by Reviewer 6uSg: “The authors re-implement 5 methods for DAOD, which proves the possibility of using this framework for future work.”  We are encouraged that reviewers see the potential for our framework to benefit future research in DAOD.

**2. A fair benchmarking protocol for DAOD methods.** We utilize our framework to re-examine flawed benchmarking protocols in DAOD and propose a more principled way forward. Reviewer vcow notes that “By isolating adaptation-specific techniques to create fairer baselines, [ALDI] clearly exposes the incremental progress of previous methods. I believe such results help the community to re-think the commonly adopted methodologies and review criteria in this field.” Similarly, Reviewer kcu1 notes ‘The authors devote significant attention to constructing a fair baseline model by comparing source-only and target-supervised methods. This step is often overlooked yet critical, as huge improvements can be observed from inadequate baselines.”

**3. A new benchmarking dataset, CFC-DAOD.** Our benchmark helps to diversify DAOD benchmarks, which at present are largely constrained to urban driving scenarios. As Reviewer vcow notes, “The new dataset proposed by this paper is indeed novel and enhances benchmarking diversity.” Our experiments demonstrate that CFC-DAOD provides a unique point of comparison that can elucidate differences between methods’ performance.

**4. A state-of-the-art DAOD method, ALDI++.** Reviewers agree that our method and experimental results are “convincing” (Reviewer vcow) and “supported by extensive experiments” (Reviwer 6uSg), and Reviewer kcu1 notes that “the paper provides a wide-ranging ablation study for the ALDI++ model […] This thorough investigation reveals valuable insights into the framework’s design and is comprehensive. Thanks to additional constructive feedback (noted below), our revision also includes additional experiments to show that ALDI++ is state-of-the-art on Faster R-CNN, YOLO, and DETR-based detection architectures.

---

> ### Author Response · Authors · 2025-01-16
> **All requested revisions addressed (cont.)**
>
> Reviewers also provided detailed and clear constructive feedback, and we are grateful for all requested revisions as they have helped improve the quality of the manuscript. The key changes we have made that concern multiple reviewers are the following:
>
> **Clarification around "modern architectures’" with respect to our main results.** We have added clarification around our choice of Faster R-CNN architectures with Resnet-50 FPN backbones for our main experiments. Methods built on top of Faster R-CNN continue to be state-of-the-art in the field of DAOD. The methods we benchmarked (AT, MIC, PT, etc.) represent the current state-of-the-art and are all built on top of Faster R-CNN, outperforming prior work in DAOD utilizing DETR and YOLO architectures. With respect to backbones, one key observation from our paper was that existing state-of-the-art DAOD methods were consistent in their architecture choice (Faster R-CNN) but inconsistent in their backbone choice (VGG vs. ResNet), leading to questions around whether the comparisons and rankings were fair. By equivalently modernizing the backbone choice—i.e. performing comparisons between
> methods where each method uses a ResNet or ViT backbone—we are able to get a more realistic picture of how
> these methods perform.
>
> **New experiments for DETR and YOLO architectures, and ConvNeXt-L and Vit-L backbones.** Reviewers felt there would be value in additional experiments with additional architectures and backbones. We argue that our main results, using Faster R-CNN and ResNet-50 FPN as described above, are a significant contribution that is well-aligned with prior work, which typically limits itself to one choice of architecture. However, we agree with reviewers that benchmarking ALDI with other detection architectures like YOLO and DETR, as well as with stronger backbones like ConvNeXt-L and Vit-L, can provide interesting insights and open doors for further work. Thanks to the architecture-agnostic nature of our framework, we have done this, and have included experiments for ALDI-DETR, ALDI-YOLO, and ConvNeXt-L and Vit-L backbones in our revised manuscript. Notably, without any hyperparameter tuning, ALDI-DETR and ALDI-YOLO outperform all prior work in DETR and YOLO-based DAOD on the Cityscapes to Foggy Cityscapes benchmark, demonstrating the robustness of our method. We hope that these contributions can help the field propose new DAOD methods for one-stage and transformer-based architectures, which, despite their strong performance on general object detection, currently lag behind Faster R-CNN in DAOD.
>
> We have addressed additional requested revisions in our other responses to individual reviewers.
>
> We again thank the reviewers and Action Editor for helping to improve our manuscript. We look forward to any further questions and feedback.

---

### Decision · Action_Editor_mpTn · 2025-03-04

**Recommendation:** Accept with minor revision

**Comment:**

This paper delves deep into the problem of domain adaptive object detection, pointing out issues and presenting good insights with rigorous, fair benchmarking. Reviewers commended the authors’ clarity in unifying existing methods within one framework, revealing the true sources of performance gains, and demonstrating how their new fish-counting dataset can highlight inconsistencies that were overlooked by driving-centric benchmarks. They found the experiments thoroughly convincing and praised the authors’ attention to detail in ablation studies, which illustrate why certain components matter more than others.

All reviewers unanimously recommended acceptance, emphasizing the high quality and clear exposition of the work. They noted that ALDI++ achieves state-of-the-art results with a well-executed evaluation across multiple architectures and backbones, reflecting both strong methodology and the potential to inspire future research. Given these strengths, the paper will be valuable to TMLR’s audience by improving transparency and best practices in DAOD, and by offering novel empirical insights that help the community move beyond poorly established baselines.

Although the inclusion of fish counting data enhances the diversity of existing DAOD benchmarks, a possible direction for future work would be extending these efforts to other scenarios (for instance, sim-to-real generalization in robotics). This would further strengthen the paper’s contribution by exploring a broader range of real-world domains, complementing the current focus on driving scenes and underwater imagery.

**Audience:**

This work would be interesting to the broad audience in several domains, including general visual recognition, visual representation learning, transfer learning, model robustness, and reliable benchmarking practices.

**Claims And Evidence:**

Yes, the primary claims in this paper are supported by convincing empirical evidence. The authors provide a thorough evaluation across multiple domain-adaptive object detection (DAOD) benchmarks, re-implement existing methods within a single framework, and conduct a series of ablation studies to demonstrate where performance gains arise. These studies support the following claims:

1) Benchmarking pitfalls exist in prior DAOD work, including underpowered baselines and inconsistent implementations.
2) ALDI++ outperforms state-of-the-art methods by a notable margin with a unified training and evaluation protocol.

In addition, the authors have carefully ablated the impact of each design component (e.g., pseudo-labeling strategies, robust burn-in, data augmentation), lending credibility to their conclusions. The paper’s methodology and results are well-explained and methodically validated, giving the claims a solid foundation.

On the other hand, the authors claim that the introduction of CFC-DAOD "enables evaluation on diverse real-world data". While this benchmark does expose differences between methods that are less captured on the standard driving-focused benchmarks, the single focused theme (fish counting) of the benchmark raises questions whether this addition alone suffices to cover broader real-world domains, as noted by reviewer kcu1. The authors are suggested to clarify this part better.

---

> ### Author Response · Authors · 2025-03-12
> **Minor revisions and camera ready version**
>
> We thank the AE and all reviewers for their insightful and constructive feedback that have certainly helped improve the manuscript. We are very pleased that all reviewers unanimously recommended acceptance, and thank the AE for their certification recommendation.
>
> We have addressed the remaining minor revision requested and uploaded a camera ready version of the manuscript. Specifically, we have clarified the language around our dataset contribution as recommended. We emphasize that CFC-DAOD is "a step toward addressing" the issue of diversity in DAOD benchmarks, rather than "addresses"; and that it "increas[es] the diversity of available DAOD benchmarks" rather than "enables evaluation on diverse real-world data". We hope that this clarifies that we believe our benchmark is a step in the right direction, but that there is also more work to be done.